# Speeding up Tsunami Forecasting to boost Tsunami Warning in Chile

Mauricio Fuentes[1], Sebastian Arriola[2], Sebastian Riquelme[2], and Bertrand Delouis[3]

[1]Department of Geophysics, Faculty of Physical and Mathematical Sciences, University of Chile
[2]National Seismological Center, Faculty of Physical and Mathematical Sciences, University of Chile
[3]Géoazur, Université de Nice Sophia Antipolis, Observatoire de la Côte d'Azur

**Correspondence:** Mauricio Fuentes (mauricio@dgf.uchile.cl)

**Abstract.** Despite the occurrence of several large earthquakes during the last decade, Chile continues to have a great tsunami-genic potential. This arises as a consequence of the large amount of strain accumulated along a subduction zone that runs parallel to its long coast, and a distance from the trench to the coast of no more than 100 km. These conditions make it difficult to implement real-time tsunami forecasting. Chile issues local tsunami warnings based on preliminary estimations of the hypocenter location and magnitude of the seismic sources, combined with a database of pre-computed tsunami scenarios. Finite fault modeling, however, does not provide an estimation of the slip distribution before the first tsunami wave arrival, so all pre-computed tsunami scenarios assume a uniform slip distribution. We implemented a processing scheme that minimizes this time gap by assuming an elliptical slip distribution, thereby not having to wait for the more time consuming finite fault model computations.We then solve the linear shallow water equations to obtain a rapid estimation of the runup distribution in the near field. Our results show that, at a certain water depth, our linear method captures most of the complexity of the runup heights in terms of shape and amplitude when compared with a fully non-linear tsunami model. In addition, we can estimate the runup distribution in quasi-real-time as soon as the results of seismic finite fault modeling become available.

## 1 Introduction

For decades, countries exposed to coastal inundation have done a lot of work to develop their tsunami warning systems (Doi, 2003; Wächter et al., 2012). Most tsunamis are generated by large subduction earthquakes and landslides, which owing to the characteristics of the tsunami source process, places a real-time tsunami forecast out of reach. Regular earthquakes follow a scaling law that links their energy release (seismic moment) to their duration (Ide et al, 2007). For instance, a regular 8.5 Mw earthquake can last for about 2 minutes, whereas we can consider tsunami generation nearly instantaneous after the source origin time. This implies that a robust tsunami warning system must integrate several systems that monitor different potential triggers such as earthquakes and volcanoes, among others. In the case of tsunamis generated by subduction earthquakes is essential to detect and characterize the seismic source. Nowadays, the W-phase method is the preferred for accounting large

earthquakes in Chile, which provides a first moment tensor solution within 5 minutes (Riquelme et al, 2016, 2018). As a matter of fact, the regional W-phase method is running now in real time in less than 5 minutes (Zhao et al., 2017). This method is based on waveform inversion theory, therefore it is necessary to have an important number of broadband seismometers in the regional field. The implementation of this method relies on robust seismic networks. This paper tries to illustrate the possibility

of replication of these examples in other countries with tsunami threat produced by earthquakes in the near field. It is well-known, however, that tsunami heights are very sensitive to the spatial slip distribution of the seismic source (Geist, 2002; Ruiz et al., 2015). Even after having a finite fault model, the simulation of the tsunami propagation can take several hours depending on the desired level of resolution. This is the reason why the tsunami forecasts of many of the current warning systems are based on pre-computed scenarios (Reymond et al., 2012; Gusman et al., 2014; Mulia et al., 2018). Chile and Japan use this

methodology for that purpose (https://www.jma.go.jp/jma/en/News/lists/tsunamisystem2006mar.pdf). This methodology, however, ignores the complexity of the seismic source and solves only for uniform slip models. We propose a methodology applicable to near-field tsunamis triggered by earthquakes that complements the monitoring systems in operation, and helps make better decisions during and after an emergency alert.

## 2  Methodology

We can separate this problem into three main parts: 1) the estimation of a seismic source model, 2) the generation of initial conditions, and 3) the corresponding tsunami simulation. We define a computation domain around the earthquake source and the coastal areas in the near field. We use the SRTM15 bathymetric data with a 15 arcsec resolution, based on the STM30 (Becker et al., 2009).

The core idea consists in trading off some accuracy to gain speed. Within the context of tectonic tsunamis generated in the near

field we want to know the places with the maximum inundation, the extension of the inundation until it decreases to 0.5 to 1 m, and the average runup. Our model does not aim at computing a detailed inundation map with the best possible accuracy, but rather to provide a fast estimate of the main area prone to inundation relying on the W-phase CMT, currently considered one of the fastest and more accurate methods to characterize the source of large earthquake (Kanamori and Rivera, 2008).

### 2.1  Slip Distribution Model

Once a W-phase solution provides a characterization of an earthquake we use an elliptical slip distribution (Dmowska and Rice, 1986) over a region determined by applying the scaling law after Blaser et al. (2010). This serves as a preliminary estimation while seismic waves are still traveling, and later finite fault solutions are computed. This in turn allows to model the near field tsunami for every finite fault model update. The elliptic model is discretized with $n_y$ subfaults along-dip and $n_x = \left\lceil \frac{L}{W} n_y \right\rceil$, where $L$ and $W$ are the length and width of the fault plane obtained with the scaling law. After setting $n_y = 16$,

all the earthquakes cases analyzed in our study have enough resolution on the source area.

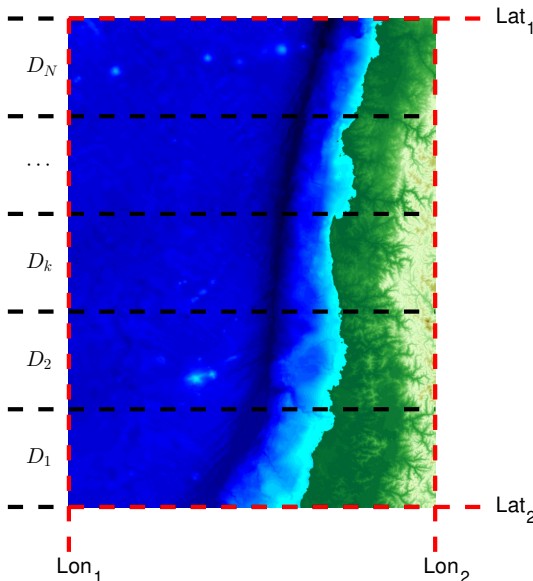

**Figure 1.** Schematic showing the discretization of the calculation domain for parallel computation.

## 2.2 Tsunami Initial Conditions

Despite evidence of influence of the source time components in the tsunami generation process, for speed purposes we model a static seafloor deformation induced by a non-uniform slip distribution that includes the horizontal components, as suggested by Tanioka and Satake (1996). This is obtained by applying the Okada equations (Okada, 1985).

## 2.3 Tsunami Modeling

The last part of this methodology is the estimation of the tsunami heights along the coast. Usually, tsunami modeling involves complex codes to solve the fully coupled non-linear shallow water equations. Depending on the domain size and resolution, a full tsunami simulation run can take several hours, which makes real-time forecast nearly impossible. To overcome this limitation, we solve the linear shallow water equations with a forward finite difference scheme. The propagation inside the domain is governed by the second order PDE with initial conditions:

$$
\begin{aligned}
\eta_{tt} - g\nabla\left(h\nabla\eta\right) &= 0 \\
\eta(x,y,0) &= \eta_0(x,y) \\
\eta_t(x,y,0) &= 0
\end{aligned}
\tag{1}
$$

where $\eta(x,y,t)$ denotes the water surface, $g$ the acceleration of gravity, $h(x,y)$ the bathymetry and $\eta_0(x,y)$ the initial condition. In the open boundaries, we set a radiation condition (Reid and Bodine, 1968), whereas in the solid boundaries

(coasts) we impose full reflection in a vertical wall placed at the 100 m isobath, before reaching the non-linear zone. Here, a Neumann boundary condition is applied: $\frac{\partial \eta}{\partial \hat{n}}$ where $\hat{n}$ denotes the exterior unit normal vector. The linear method (LM) allows to obtain a faster estimation than a full tsunami code since second-order terms are disregarded while still accounting for the same main features. In addition, this approach does not require computing the velocity field, an added benefit that makes the computation programs even faster. Each simulation is compared to its corresponding full non-linear shallow water equation propagation. We use the JAGURS code (Baba et al., 2014) written in Fortran90 using parallel arrays via OpenMP and OpenMP + MPI. This code is based on the classic finite difference method of Satake (2002). For each scenario, we run the simulation for the equivalent of two hours of tsunami travel time to obtain the main features of the runup distributions, despite the fact that later amplification of edge waves and resonances effects can occur. The approximated runup is obtained as the maximum from the vertical wall reflexion boundary condition. The resulting runup values are in the same order of the actual runup for a sloping beach model (Synolakis, 1987).

## 3  Implementation and Benchmarking

To evaluate the performance of our approach, we modeled nearly all the great tsunamis of the last two decades. Most of them were already tested with an analytical approach in (Riquelme et al, 2015). The details of the propagation and runup distribution of the 12 events tested are presented in the supporting information. For these examples we used the finite fault models provided by the USGS (Hayes, 2017), as they have proven operationally robust for real time operations in the context of global monitoring. All the computations were performed in a Dell Precision 7920, with two Intel Xeon Gold 6136 processors, each with 12 physical cores, for a total of 24 physical cores, and 2 threads each. For each time iteration, the domain is divided into 48 subdomains that are computed in different threads, for a parallel array (Fig.1). To compute the tsunami initial condition, the Okada equations were implemented in the C programming language using threading, together with the finite difference scheme for the LM. The C code uses the pthread library to define a C data structure containing a pthread, and then calls a function that sends each grid subdomain to threads running in different cores for computation. This method is as reliable as any other linear scheme method, as it solves the same equations. The only significant difference is in the threads distribution for time optimization. When a thread finishes, it computes for a certain time step and it joins with the others in order to avoid miscomputations. For instance, on the system used to run our computations for a regular grid of 4 million points with an FFM of 300 subfaults, the vertical and horizontal seafloor displacements can be calculated almost instantly (less than 5 seconds), and two hours of tsunami wave propagation for the 2011 Tohoku, Japan earthquake can be solved in 60 seconds.

## 4  Discussion of Computational Results

All the earthquakes presented here, have produced tsunamis. The range of magnitude varies from 7.7 to 9.1. They occurred in different subduction zones around the world. The largest ones are Tohoku in Japan and Maule in Chile. All of them show a thrust mechanism except for the Samoa event in 2009 which is a normal event. There are a few tsunami earthquakes in this

section such as the 1992 Mw 7.7 Nicaragua Earthquake, The 2006 Mw 7.6 Java Earthquake. The geometry of the earthquakes causative fault varies from $L = 150$ km to $L = 500$ km, the range of peak displacement at the source varies from 3 m to 40 m. Therefore, we have tested as many earthquakes and as many source features as possible for this study.

1. The 1992 Mw 7.7 Nicaragua Tsunami Earthquake

2. The 2001 Mw 8.4 Southern Perú Earthquake

3. The 2003 Mw 8.3 Hokkaido Earthquake

4. The 2006 Mw 7.6 Java Earthquake

5. The 2007 Mw 8.1 Solomon Islands Earthquake

6. The 2007 Mw 8.0 Pisco Earthquake

7. The 2009 Mw 8.1 Samoa Islands Region Earthquake

8. The 2010 Mw 8.8 Maule Earthquake

9. The 2011 Mw 9.0 Tohoku Earthquake

10. The 2012 Mw 7.8 British Columbia Earthquake

11. The 2014 Mw 8.2 Iquique Earthquake

12. The 2015 Mw 8.3 Illapel Earthquake

For each event we apply the methodology previously described, and use the W-phase centroid moment tensor, a scaling law, and an elliptic slip distribution to define the first source. Then, the linear and non-linear tsunami simulations are performed. The resulting runup distributions are decomposed along latitude and longitude in order to compare both models. The same procedure is repeated, this time considering an FFM solution instead. Table 1 shows the correlation between the runup distributions obtained with the JAGURS code (non-linear method) and the method presented in this paper (linear method). Table 2 summarizes the CPU times in seconds for different stages of the process for each simulation. There is a high degree of agreement within a short time. Detailed figures showing the results for the 24 simulations are provided in the supplementary material, where maximum amplitudes, runup distribution, and field measurements are listed. For comparison purposes, the event of 2014 Chile, the DART station 32401 registered 0.25 m of amplitude (An et al., 2014), where the linear method predicts 0.39 m for the elliptic source and 0.12 m for the FFM, whereas JAGURS gives 0.55 m for the elliptic source and 0.15 m for the FFM.

**Table 1.** Correlation of the runup distribution obtained from our linear model solution and the JAGURS code. Correlation is computed with the standard Pearson coefficient. Details can be found in the supplemental material.

| Event | FFM lon | FFM lat | Elliptic lon | Elliptic lat |
|---|---|---|---|---|
| 1992 Nicaragua | 0.8323 | 0.8088 | 0.8841 | 0.8587 |
| 2001 Perú | 0.8334 | 0.8575 | 0.6697 | 0.7549 |
| 2003 Japan | 0.7753 | 0.7838 | 0.9139 | 0.9129 |
| 2006 Indonesia | 0.7483 | 0.8531 | 0.8134 | 0.9030 |
| 2007 Solomon Isl. | 0.6422 | 0.7575 | 0.8412 | 0.8626 |
| 2007 Perú | 0.8380 | 0.8085 | 0.8872 | 0.8896 |
| 2009 Samoa | 0.6987 | 0.7353 | 0.7779 | 0.8093 |
| 2010 Chile | 0.7346 | 0.6039 | 0.8682 | 0.7820 |
| 2011 Japan | 0.8571 | 0.7074 | 0.9229 | 0.8311 |
| 2012 Canada | 0.6829 | 0.6034 | 0.8731 | 0.8398 |
| 2014 Chile | 0.7833 | 0.6473 | 0.9051 | 0.8341 |
| 2015 Chile | 0.9103 | 0.7663 | 0.9603 | 0.8686 |

**Table 2.** Summary of the CPU time in seconds for the twelve events. $t_{IC}$ indicates the time needed to compute the initial conditions, $t_{Pr}$ the processing time, $t_{TP}$ the time to compute the tsunami propagation, and $t_T$ the total time.

| Event | FFM | | Elliptic | | LM | JAGURS | Total time $t_T$ | | | |
|---|---|---|---|---|---|---|---|---|---|---|
| | $t_{IC}$ | $t_{Pr}$ | $t_{IC}$ | $t_{Pr}$ | $t_{TP}$ | $t_{TP}$ | FFM-LM | FFM-JAGURS | Elliptic-LM | Elliptic-JAGURS |
| 1992 Nicaragua | 6 | 4 | 8 | 5 | 31 | 575 | 41 | 585 | 44 | 586 |
| 2001 Perú | 5 | 3 | 7 | 3 | 18 | 360 | 26 | 368 | 28 | 370 |
| 2003 Japan | 5 | 3 | 9 | 3 | 22 | 428 | 30 | 436 | 34 | 440 |
| 2006 Indonesia | 4 | 3 | 7 | 2 | 20 | 358 | 27 | 365 | 29 | 367 |
| 2007 Solomon Isl. | 8 | 5 | 10 | 5 | 28 | 658 | 41 | 671 | 43 | 673 |
| 2007 Perú | 4 | 4 | 9 | 4 | 31 | 546 | 39 | 554 | 44 | 559 |
| 2009 Samoa | 4 | 3 | 6 | 2 | 17 | 321 | 24 | 328 | 25 | 329 |
| 2010 Chile | 7 | 4 | 9 | 5 | 32 | 651 | 43 | 662 | 46 | 665 |
| 2011 Japan | 6 | 6 | 13 | 6 | 46 | 223 | 58 | 235 | 65 | 242 |
| 2012 Canada | 2 | 1 | 4 | 1 | 15 | 153 | 18 | 156 | 20 | 158 |
| 2014 Chile | 5 | 4 | 11 | 5 | 24 | 500 | 33 | 509 | 40 | 516 |
| 2015 Chile | 4 | 4 | 8 | 4 | 27 | 500 | 35 | 508 | 39 | 512 |

## 5 Application to compliment tsunami alert. Case study: The 2015 Illapel Earthquake

On 16 September, 2017 an 8.3 Mw earthquake occurred in the Coquimbo region, Chile (Melgar et al., 2016; Fuentes et al., 2017). The characteristics of this event made it an ideal case study for tsunami generation. The national agencies implemented the established protocols for evacuating the whole Chilean coast, even the more distant insular territories (SNAM, bulletin 1, Sept. 16th, 23:02 5 UTC). Such decisions have to be made within minutes of origin time. In general, an accurate prediction of the tsunami runup heights requires a precise image of the seismic source, which at present is not available within 5 minutes for real-time after adding the tsunami simulation times. Nevertheless, we can come close to a quasi-real-time approach by triggering a first estimation assuming an elliptical slip distribution. This only takes a few seconds, and can at present be done instead of searching a pre-computed database of scenarios that are usually limited. For monitoring purposes, the results can be updated everytime a seismic source imaging is received, for both, the near field (at 15 arcsec) and regional field (at 60 arcsec). All this information is summarized in a color-coded map following the official coding used by the Chilean institutions (Melgar et al., 2016). Color coded maps are self-explanatory, which makes them easy to interpret (Figs.2 and 3). Each region can then be rapidly assigned a color linked to an specific evacuation protocol. All the simulations were performed for two hours of tsunami propagation where the main energy content plays a key role on the inundation process. Figure 4 illustrates the normalized energy rate that generates the runup history along the coast, showing that the majority of the global energy is concentrated within the first hour. We can also observe that the first estimation obtained for an elliptical fault predicts the same levels of inundation as the full finite fault model in the near field, while we can observe minor differences in the regional field. This makes sense, since finite fault model results become available during the tsunami monitoring stage, when time is not as critical as in the very first minutes after origin time. It has to be noticed that is possible to increase the number of warning levels allowing to find the optimal number of states for emitting and communicating the warning bulletin. In this study we choose the UNESCO standard. For completeness, we computed the travel time isochrones across the Pacific Basin (Fig. 5). These computations use a dense set of rays following Sandanbata et al. (2018), which allows to include dispersive effects. We have also included the effect of the earth elasticity as shown in An and Liu (2016). These kind of maps can be computed instantly together with the very first estimation of the moment tensor and then updated.

## 6 Conclusions

In this study we propose a method that disregards the fine complexity of the seismic source while using fine bathymetric data and a set of simplified equations. Implementation of this method allows to model more than 80% of the tsunami runups with enough accuracy for tsunami warning purposes up to 20 times faster. Our method also aims at rapidly predicting the spatial distribution of the tsunami runups using some simplifications in the tsunami equations. Despite lacking the mathematical rigorosity that we would otherwise prefer, the method we propose is not inexact within the context of an emergency response system that needs to trigger actions that can potentially save lives and reduce economic losses after the occurrence of a large earthquake. We summarized our approach in the flowchart shown in Figure 6. Taking into account the results of our study we can list the following as the most noteworthy results:

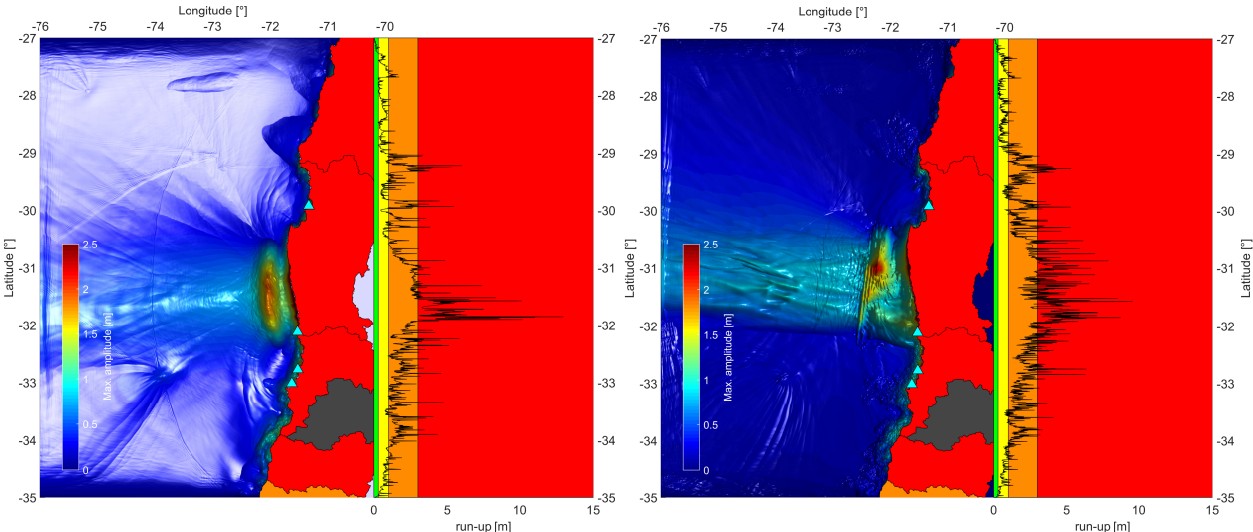

**Figure 2.** Near field simulation of the 2015 Illapel earthquake with an elliptical source (left), and a finite fault model (right). The colors assigned to different areas indicate the expected runups in meters: a) red for runups larger than 3 m, b) orange for runups between 1 and 3 m, c) yellow for runups between 0.3 and 1 m, and d) green for runups smaller than 0.3 m.

1. Although other tsunami warning centers use linear theory as part of their operations, for instance at the PTWC (http://unesdoc.unesco.org/images/0022/002203/220368e.pdf), in this study we have combined it with the use of more complex sources and faster algorithms to generate a unique and simple product easy to interpret.

2. The non-complexity of the adopted source does not seem to significantly affect the results of a fast tsunami runup estimation for emergency response purposes. By computing different levels of tsunami hazard in near-real time we can estimate more accurately the extent of the area potentially affected by the tsunami, the maximum level of inundation, and how many people will be exposed to this hazard along the Chilean coast.

3. Using the methodology of Sandanbata et al. (2018) it is possible to instantaneously calculate the tsunami arrival times from sources generated in the far field with enough accuracy. This can also be done via tsunami modeling, but at the expense of longer computation times.

4. When compared to other tsunami modeling codes such as JAGURS, results obtained from our method match more than 80% of the predicted runup for 15 arcsec bathymetry while obtaining the results at least 20 times faster.

5. The simple method proposed in this study provides a fast, reliable, and intuitive characterization of the tsunami threat, which in turn allows disaster mitigation agencies to take appropriate action.

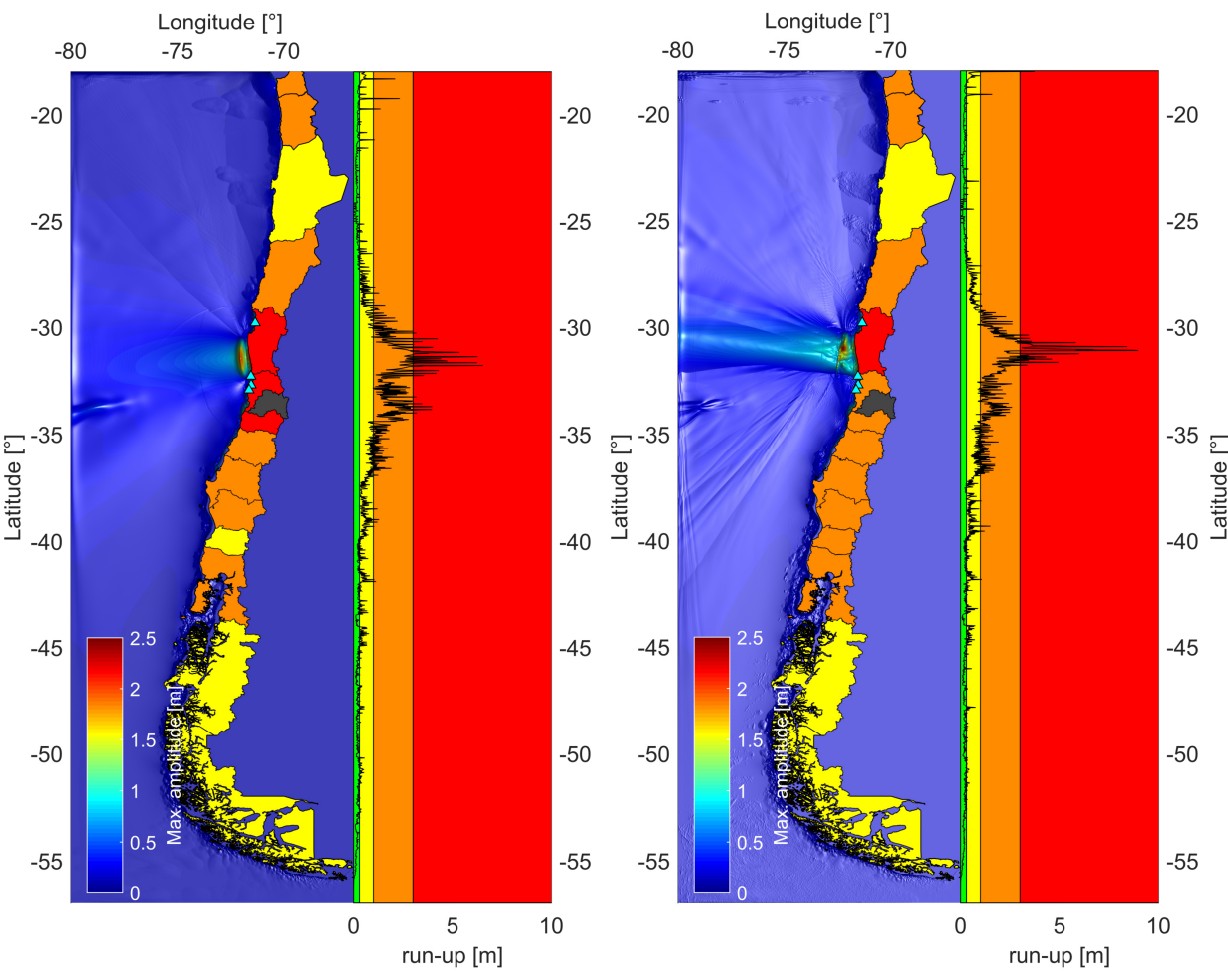

**Figure 3.** Regional field simulation of the 2015 Illapel earthquake for an elliptical source (left), and a finite fault model (right). The colors assigned to different areas indicate the expected runups in meters: a) red for runups larger than 3 m, b) orange for runups between 1 and 3 m, c ) yellow for runups between 0.3 and 1 m, and d) green for runups smaller than 0.3 m.

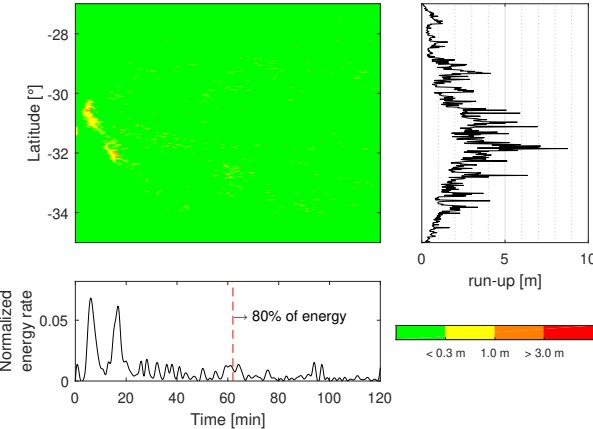

**Figure 4.** Normalized runup energy rate during the first two hours of tsunami simulation. The upper left panel shows the runup rate along latitude and time, the upper right panel shows the final maximum runup, and the bottom left panel shows the normalized energy rate for the whole process as a time series.

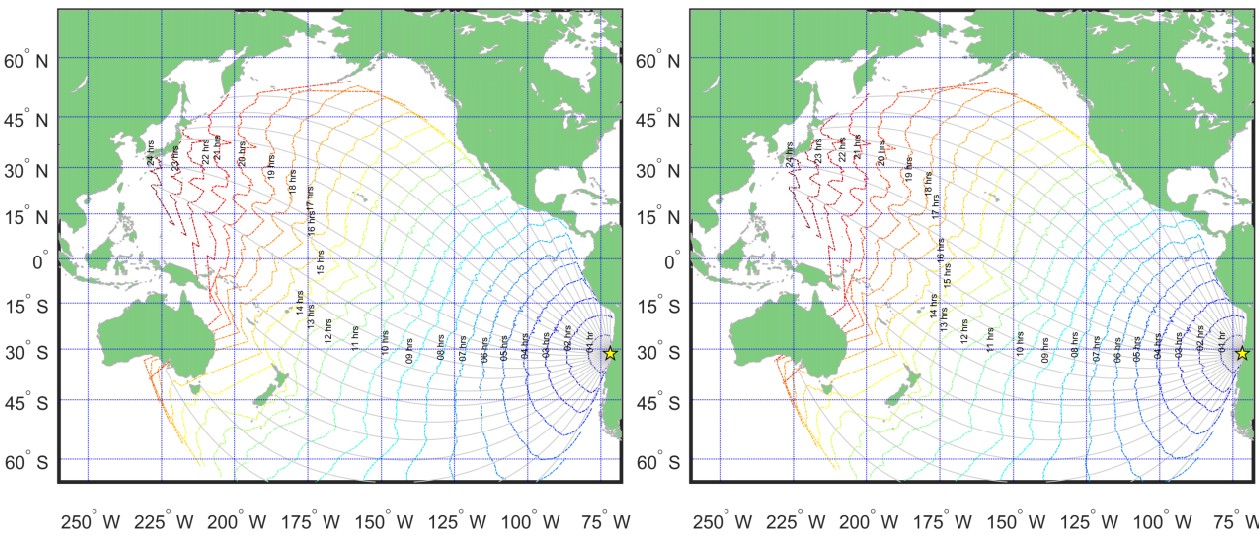

**Figure 5.** Tsunami travel times across the Pacific basin for the 2015 Illapel earthquake. The left panel shows the travel times after the shallow water equations, while the travel times in the right panel include the effects of dispersion and the earth elasticity for a wave frequency of 2 mHz.

*Acknowledgements.* This study was enterally supported by the *Programa de Riesgo Sśmico*.

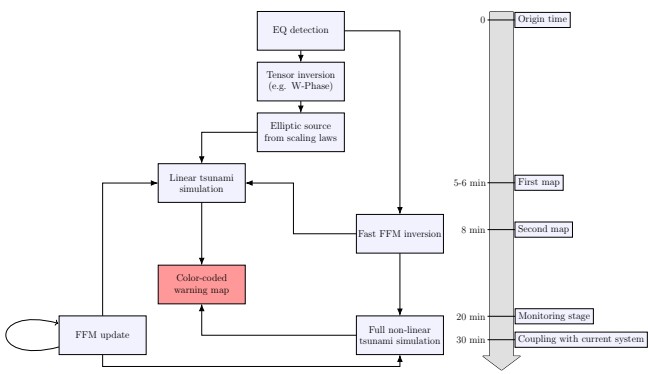

**Figure 6.** Flow chart of the methodology proposed in this study.

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
