# Peer review of "Speeding up Tsunami Forecasting to boost Tsunami Warning in Chile"

_Natural Hazards and Earth System Sciences, 2019_

## Referee Comment (RC1) · Anonymous Referee #1 · 4 Mar 2019

The paper presents a methodology aimed at speeding up the generation of a tsunami forecast as part of tsunami warning operations. The authors modeled tsunamis for twelve of the largest earthquakes that occurred between 1992 and 2015 applying a) their newly proposed linear method assuming an elliptical slip distribution, and b) a fully non-linear method. Comparison of the results indicates that the proposed linear method allows the generation of a much faster tsunami forecast that matches the results of the fully non-linear method with an accuracy of up to 80% but 20 times faster. These results make the paper worthy of publication and of interest to the tsunami warning and disaster management community. As written, however, the paper needs a significant amount of work before we can consider it ready for publication. The text needs major revisions to improve its overall readability and flow. Instead of providing

an exhaustive list of all the grammar and composition issues we have found, however, we have taken the liberty of editing most of the text and will attach it as suggested edits with the hope that it will help with the editing process. Please find below some additional comments and specific suggestions:

1) The title of the paper does not reflect the actual contents and results presented in the paper.

The title of the paper suggests that the research results included in it will speed up the issuance of tsunami warnings in Chile. At present, however, most tsunami warning protocols implemented in the world rely on using a quick estimate of an earthquake's magnitude as a proxy to evaluate its tsunamigenic potential. To date, within the context of tsunami warning operations, only the P-wave moment magnitude (Mwp) method implemented at the US Tsunami Warning Centers in the late nineties has significantly sped up tsunami warning in general. More recent earthquake magnitude estimations methods like W-phase, although more robust, accurate, theoretically sound, and faster than other CMT methods, lacked and still lack the speed needed to truly speed up tsunami warning in general. At the time of publication of the seminal paper on the Wphase CMT method paper in 2008, for instance, the PTWC routinely issued tsunami warnings and tsunami messages within 12 minutes of origin time. At present, the PTWC issues its tsunami message products, on average, in less than 6 minutes of origin. In contrast, it still takes between 20∼25 minutes to obtain the results of a W-phase CMT inversion, and around 10∼15 minutes for a regional implementation. Faster implementations turn possible only in regions with a high density of seismic stations like Chile, Japan, or the West Coast of the United States. Even for these regions the generation an issuance of a tsunami message in less than 5∼6 minutes turns close to impossible relying on a W-phase solution. In other words, despite the paper's title, the proposed linear tsunami simulation methodology does not speed up the issuance of tsunami warnings in Chile. The proposed linear method seems to rather speed up considerably the generation of tsunami propagation and inundation forecasts that provide faster and more accurate

estimates than those currently in operation. For this reasons, the authors should consider changing the title of the paper to something more reflective of both, the scope of the paper and its results such as: "Speeding up Tsunami Forecasting to boost Tsunami Warning in Chile", with the possible substitution of "boost" with "enhance" or "improve" instead.

b) Towards the end of the introduction the authors state that many of the current warning systems have pre-computed tsunami scenarios at their core. This turns inaccurate, as most warning systems currently operational in the world use the preliminary earthquake location and magnitude as a proxy to evaluate tsunamigenic potential and issue their warnings accordingly. Many use pre-computed tsunami scenarios to generate a tsunami forecast following that initial warning, while others use a combination of pre-computed tsunami scenarios and real time tsunami simulations based on the linear shallow water equations. Generation of this last type of forecast currently takes between 3 to 7 seconds for an area covering 1000 to 1500 square kilometers around the earthquake's epicenter, and 10 minutes or less for the whole Pacific basin depending on magnitude and resolution settings.See the reviewed text in the pdf file for suggested edits.

b) All twelve historical earthquakes used in the study generated tsunamis recorded by sea-level instruments, either by tide gauges located along the coast or by DART buoys located in deep water. The paper would benefit by the inclusion of a table listing the tsunami waves heights recorded at these point locations together with the corresponding wave heights predicted by both the linear, and non-linear modeling approaches. Doing so would validate not only a model against another considered theoretically superior but also against the actual field measurements of the phenomenon under study. This turns into the ultimate validation of the accuracy and usefulness of both tsunami modeling, and any forecast based on it.

c) The authors should consider renaming some of the sections as suggested in the attached pdf file. In addition, the conclusions should list the most relevant results of the

study after a brief summary of the work done in the paper. We attempted to summarize the results in the attached pdf file containing a reviewed version of the text, but the authors should consider adding or modifying whatever they consider relevant.

d) The labels of figures and tables should describe their contents to make them self-contained. When referencing the figures inside the text we suggest applying the same format to all instances, as for instance "Fig. 1", or "Figs. 2 and 3" instead of using "Figure 1". Please find below a list of suggested edits to the current labels of Figures and Tables in the main text. Consider applying similar edits to the labels included in the supplement:

Figure 1. Schematic showing the discretization of the calculation domain for parallel computation.

Figure 2. Near field simulation of the 2015 Illapel earthquake with an elliptical source (left), and a finite fault model (right). The colors assigned to different areas indicate the expected run-ups in meters: a) red for run-ups larger than 3 m, b) orange for run-ups between 1 and 3 m, c ) yellow for run-ups between 0.3 and 1 m, and d) green for run-ups smaller than 0.3 m.

Figure 3. Regional field simulation of the 2015 Illapel earthquake for an elliptical source (left), and a finite fault model (right). The colors assigned to different areas indicate the expected run-ups in meters: a) red for run-ups larger than 3 m, b) orange for run-ups between 1 and 3 m, c ) yellow for run-ups between 0.3 and 1 m, and d) green for run-ups smaller than 0.3 m.

Figure 4. Normalized run-up energy rate during the first two hours of tsunami simulation. The upper left panel shows the run-up rate along latitude and time, the upper right panel the final maximum run-up, and the bottom left panel the normalized energy rate for the whole process as a time series.

Figure 5. Tsunami travel times across the Pacific basin for the 2015 Illapel earthquake.

[Figure]

The left panel shows the travel times after the shallow water equations, while the travel times in the right panel include the effects of dispersion and the earth elasticity for a wave frequency of 2 mHz.

Figure 6. Flow chart of the methodology proposed in this study.

Table 1. Correlation of the run-up distribution obtained from our linear model solution and the JAGURS code.

Table 2. Summary of the CPU time in seconds for the twelve events. tIC indicates the time needed to compute the initial conditions, tPr the processing time, tTP the time to compute the tsunami propagation, and tT the total time.

Please also note the supplement to this comment:
https://www.nat-hazards-earth-syst-sci-discuss.net/nhess-2019-9/nhess-2019-9-RC1-supplement.pdf

―――――――――――――――――――

[Figure]

**Supplement:**

[revised manuscript text omitted]
 and had moment magnitudes between 7.7 to 9.1. Among them the 2011 Tohoku, Japan, and 2010 Maule, Chile earthquakes had the largest magnitudes. All these events had a thrust fault mechanism, except for the Samoa event in 2009, which had a normal fault mechanism. Both the 1992 Mw 7.7 Nicaragua earthquake, and the 2006 Mw 7.6 Java earthquake classify as tsunami (slow) earthquakes. The length (L) of the causative faults varies from 150 to 500 km, while the range of the peak displacements at the source varies from 3 to 40 meters. This variety aims at testing our approach with as many earthquakes and as many source features as possible:

1. The 1992 Mw 7.7 Nicaragua tsunami earthquake

2. The 2001 Mw 8.4 Southern Perú earthquake3.

3. The 2003 Mw 8.3 Hokkaido earthquake

4. The 2006 Mw 7.6 Java, Indonesia earthquake

5. The 2007 Mw 8.1 Solomon Islands earthquake

6. The 2007 Mw 8.0 Pisco, Perú earthquake

7. The 2009 Mw 8.1 Samoa Islands Region earthquake

8. The 2010 Mw 8.8 Maule earthquake

9. The 2011 Mw 9.0 Tohoku earthquake

10. The 2012 Mw 7.8 British Columbia earthquake

11. The 2014 Mw 8.2 Iquique, Chile earthquake

12. The 2015 Mw 8.3 Illapel, Chile earthquake

For each event we apply the methodology previosly described, and use the W-phase centroid moment tensor, a scaling law, and a elliptic slip distribution to define the first source. Then, the linear and non-linear tsunami simulations are performed. The resulting run-up distributions are decomposed along latitude and longitude in order to compare both models. The same procedure is repeated, this time considering a FFM solution instead. Table 1 shows the correlation between the runup distributions obtained with the JAGURS code (non-linear method) and the method presented in this paper (linear method). Table 2 summarizes the CPU times in seconds for different stages of the process for each simulation. There is a high degree of agreement within a short time. Detailed figures showing the results for the 24 simulations are provided in the supplementary material, where maximum amplitudes, runup distribution, and field measurements are listed.

**5 Application of the Method to Complement Tsunami Warning: The 2015 Illapel Earthquake Case Study**

On 16 September, 2017 an 8.3 Mw earthquake occurred in the Coquimbo region, Chile (Melgar et al., 2016; Fuentes et al., 2017). The characteristics of this event made it an ideal case study for tsunami generation. The national agencies implemented the established protocols for evacuating the whole Chilean coast, even the more distant insular territories (SNAM, bulletin 1, Sept. 16th, 23:02 UTC). Such decisions have to be made within minutes of origin time. In general, an accurate prediction of the tsunami run-up heights requires a precise image of the seismic source, which at present is not available within 5 minutes or more for real-time after adding the tsunami simulation times. Nevertheless, we can come close to a quasi-real-time approach by triggering a first estimation assuming an elliptical slip distribution. This only take a few seconds, and can at present be done instead of searching a pre-computed database of scenarios that are usually restricted. For monitoring purposes, the results can be updated everytime a seismic source imaging is received, for both, the near field (at 15 arcsec) and regional field (at 60 arcsec). All this information is summarized in a color-coded map following the official coding used by the Chilean institutions (Melgar et al.,2016). Color coded maps are self-explanatory, which makes them easy to interpret (Figs. 2 and 3). Each region can then be rapidly assigned a color linked to an specific evacuation protocol. All the simulations were performed for two hours of tsunami propagation where the main energy content plays a key role on the inundation process. Figure 4 illustrates the normalized energy rate that generates the runup history along the coast, showing that the majority of the global energy is concentrated within the first hour. We can also observe that the first estimation obtained for an elliptical fault predicts the same levels of inundation as the full finite fault model in the near field, while we can observe minor differences in the regional field. This makes sense since finite fault model results become available during the tsunami monitoring stage, when time is not as critical as in the very first minutes after origin time. For completeness, we computed the travel time isochrones across the Pacific Basin (Fig. 5). These computations use a dense set of rays following Sandanbata et al. (2018), which allows to include dispersive effects. We have also included the effect of the earth elasticity as shown in An and Liu (2016). This kind of maps can be computed instantly together with the very first estimation of the moment tensor and then update them.

**6 Conclusions**

In this study we propose a method that disregards the fine complexity of the seismic source while using fine bathymetric data and a set of simplified equations to model more than 80% of the runups with enough accuracy for tsunami warning purposes up to 20 times faster. Our method also aims at rapidly predicting the spatial distribution of the run-ups using some simplifications in the tsunami equations. Despite lacking the mathematical rigorosity that we would otherwise prefer, the method we propose is not inexact within the context of an emergency response system that needs to trigger actions that can potentially save lives and reduce economic losses after the occurrence of a large earthquake. Taking into account the results of our study we can list the following as the most noteworthy results:

1. Although other tsunami warning centers use linear theory as part of their operations, for instance at the PTWC (http://unesdoc.unesco. in this study we have combined it with the use of more complex sources and faster algorithms to generate a unique and simple product easy to interpret.

2. The non-complexity of the adopted source does not seem to significantly affect the results of a fast run-up estimation for emergency response purposes. By computing different levels of tsunami hazard in near-real time we estimate more accurately the extent of the area potentially affected by the tsunami, the maximum level of inundation, and how many people will be exposed to this hazard along the Chilean coast.

3. Using the methodology of Sandabata el al. it is possible to instantaneously calculate the tsunami arrival times from sources generated in the far field with enough accuracy. This can also be done via tsunami modeling, but at the expense of longer computation times.

4. When compared to other tsunami modeling codes such as JAGURS, results obtained from our method match more than 80% of the predicted runup for 15 arcsec bathymetry while obtaining the results at least 20 times faster.

5. The simple method proposed in this study provides a fast, reliable, and intuitive characterization of the tsunami threat, which in turn allows disaster mitigation agencies to take appropriate action.

*Code availability.* TEXT

*Data availability.* TEXT

*Code and data availability.* TEXT

*Sample availability.* TEXT

*Video supplement.* TEXT

**Appendix A**

**A1**

*Author contributions.* TEXT

*Competing interests.* TEXT

5   *Disclaimer.* TEXT

*Acknowledgements.* TEXT

**References**

REFERENCE 1

REFERENCE 2

---

## Referee Comment (RC2) · Anonymous Referee #2 · 6 Mar 2019

General comments: The paper discusses a rapid estimation of potential tsunami energy distribution along the coast (or at certain isobath) from any given earthquake parameters. As a typical forecasting algorithm, the main issue here is the tradeoff between the speed and accuracy. To obtain a timely warning, the proposed algorithm uses a rough source estimate from the W-phase inversion as well as a linear tsunami model. Despite the simplifications, the model produces a sufficient level of accuracy to facilitate the early warning system. Additionally, the proposed method is rigorously tested against historical tsunami events, which is another important factor of this paper that make it worthy of publication. In general, the paper is well-written (except for the discussion and conclusions section) and the main message to convey is easy to follow. However, I would recommend further clarifications in some parts, which can be found

in the following specific comments, before the paper can be accepted for publication.

Specific comments:

1. Page 1 Line 8. "Our results show that ... non-linear tsunami code." The sentence can be misleading. I would suggest to revise it into "Our results show that, at a certain water depth, this linear method ...".

2. Page 2 Line 1. "... are based on precomputed scenarios". Adding a sentence here with reference to the previous related works would better justify the statement. Here are some papers that can be considered:

Reymond, D., Okal, E. A., Hébert, H., & Bourdet, M. (2012). Rapid forecast of tsunami wave heights from a database of pre-computed simulations, and application during the 2011 Tohoku tsunami in French Polynesia. Geophysical Research Letters, 39(11).

Gusman, A.R., Tanioka, Y., MacInnes, B.T. & Tsushima, H., 2014. A methodology for near-field tsunami inundation forecasting: Application to the 2011 Tohoku tsunami, J. geophys. Res.: Solid Earth, 119(11), 8186–8206.

Mulia, I. E., Gusman, A. R., & Satake, K., 2018. Alternative to non-linear model for simulating tsunami inundation in real-time. Geophysical Journal International, 214(3), 2002-2013.

3. Page 3 Equation 1. The first line of the equations is a linearized SWE, and the second line refers to the initial condition. What about the third line? Derivative of elevations with respect to time at $t = 0$?

4. Page 3 Line 19. The tsunami propagation is limited at 100 m isobath, while in the supporting information the simulated runups are compared with the actual runup observations. I am aware that the paper aims to estimate the possible runup distribution in general by disregarding the physic of nearshore processes due to the nature of the algorithm. However, such an inconsistent comparison needs to be clearly defined. For example, by including the 100 m water depth contour on the plots and additional sentences in the figures caption explaining about the difference of runup locations between observation and model. Or, better yet, why don't use Green's Law as in the Raymond et al. (2012)?

5. Page 5 Table 1. The use of "lon" and "lat" is rather confusing without seeing the corresponding figures. I suggest to add a reference to the supporting information in the Table caption, though it has been mentioned somewhere in the text. Furthermore, mathematical formulation of the correlation coefficient can also be a good addition for the supporting information.

6. Page 6 Table 2. Please explain why the computational time of the elliptic slip distribution is longer than the FFM? From the figures in the supporting information I can see that the elliptic slip models have a smaller subfault size. If that is the case, information on the subfault size should be added in section 2.1 including the reasons for using finer resolution in the elliptic slip model. Also, tT in the caption is written tR on the table.

7. Page 6 Line 15. It is difficult to grasp the meaning of the last sentence. Please rewrite it.

8. Page 7. The flow of descriptions in the discussion and conclusions section is not very smooth. Improvements can be done by either rewriting the whole paragraphs or using bullet-points or numbers to indicate different topic of discussion.

9. For the Java case (Figure S7), it seems like the fault of the elliptic source is located seaward of the trench (in the outer rise region). If this is true, then the model needs to be revised, because the 2006 West Java event was a shallow interplate earthquake (a typical tsunami earthquake), which is better depicted in the FFM solution (Figure S8). Other than that, the Java Island map in the left panel is inaccurate. I believe this may be caused by a wrong color map scale used for plotting. Please also check the other locations.

2019-9, 2019.

---

## Referee Comment (RC3) · Anonymous Referee #3 · 20 Mar 2019

This paper presents a method for quick tsunami estimation in Chile early warning system using W-shape inversion for rough source estimation and linear tsunami numerical modeling. They mention this new approach as a fill-in gap method for the warning system. They have also tested their method with historical tsunami events and proved that they have good correlation between the real case and their results. This work is worth to be published. However, it needs major revision in terms of presenting and discussing their results, grammar in the entire text and conclusion. Besides, it is necessary to have further discussion and explanation on figures.

Below are the sentences that are not clear to me. They need to be rephrased: 1. Page 1 Line 14 whole sentence 2. Page 2 Line 4 "operating monitoring systems" 3. Page 2 Line 11 "until decreases to 0.5 to 1 m" 4. Page 5 Line 11 . . . whole paragraph 5. Page

5 last paragraph 6. Page 6 Line 4 to 6 7. Page 9 last paragraph

Below are some of grammar corrections: 1. Page 1 Line 3 "100 km which creates" 2. Page 2 Line 6 "This problem is separated in three parts: the determination of..." 3. Page 3 Line 24 "...with a fully linear shallow water equation propagation." 4. Page 4 Line 24 "....have tested as many earthquakes listed below and..." 5. Page 6 Line 2 "...very first minutes..." 6. Page 6 Line 11 "...which deploy a specific..." 7. Page 7 Line 1 "These kind of maps..." 8. Page 7 Line 5 "with a unique and simple..." 9. Page 7 Line 9 "...and number of people exposed to this hazard..." 10. Page 7 Line 13 "...details rather than modeling..." 11. Page 10 Line 1 "Using the methodology of Sandabata el al. (2018)..."

About Figures and Tables: 1. Please refer to Figure 1 in the text. 2. please insert Table 1 after its reference in the text. 3. Figure 2: Does this red color on land represent runup values larger than 3m? Does this mean all of this are experienced more than 3m runup?? 4. Figure 5: please compare and discuss the difference between two maps: in terms of the effect of dispersion etc.. 5. Figure 6: please refer Figure 6 in the text and explain.

---

## Referee Comment (RC4) · Anonymous Referee #4 · 27 Mar 2019

The paper presents a methodology to speeding up the tsunami forecast in Chile as part of tsunami warning operations. This is a very important topic, in particular because in the last decade many new tsunami warning centers have been established by various countries. This paper presents interesting results for publication. Nevertheless several issues should be explained, discussed and many data are missing, before accepting the paper. Major revision is necessary.

The results of proposed methods depends mainly on the variation of the source parameters between the different methods used, in particular the slip, the dip and the dimension and location of rupture zone, and the focal depth. One first request : the list and values of parameters of the sources of that paper are missing.

It can already been checked on the various maps presented, that the location of the

epicenter for the elliptic model, and the location of the center of the rupture zone of the fault model are not the same, and there are not the GCMT location. Why ? How do the authors decide the location of the epicenter, and why the locations are different for the different models ?

The second question is why did the authors analyzed the results of such method along other coastlines than Chile ? It doesn't provide any results about the variability of the warning forecast along the Chilean coastlines. On the other hand, two missing recent events have not been modeled and should be added to the study: Chile 1985 and Antofagasta 1995.

One of the recent papers that describes the effectiveness and rapidity of the W-Phase to get robust centroid moment tensor solutions is by J. Roch at al. (Roch, J., Duperray, P. and Schindelé F. (2016) Very fast characterization of focal mechanism parameters through W-Phase Centroid inversion in the context of tsunami warning, Pure Appl. Geophys. 173 (2016), 3881–3893, DOI 10.1007/s00024-016-1258-3). In that paper, the authors analyzed W-Phase results at global and regional scale with specific Green's functions to provide accurate solution in 15 minutes (10 minutes of signal). Due to the characteristics of the very long period W-Phase, it wouldn't be physically feasible to compute sooner W-Phase waves.

But it is well know that the first tsunami wave could impact the Chilean coastlines in less than 15 minutes. The mandate and goal of the National tsunami warning center that is facing near-field tsunami warning is to provide the first warning message in less than 15 minutes after the quake occurrence. As the results of W-Phase would not be available, the authors should explain how they would proceed to provide this first bulletin. The authors should identify a preliminary solution to perform modeling before getting the results of the W-Phase computation, and getting results in 15 minutes after the quake. Second point, the authors informed that for their study, they used W-Phase results. How was computed the parameters of all these past events ? In particular, the location of the centroid moment tensor, and on the strike and dip values used for

the elliptic method. On this specific method, the authors should present what are the parameters of the seismic sources needed for the elliptic method, and the values of the parameters for all the events processed in this paper.

The next issue is how they plan to implement the complementary messages using W-Phase source parameters. Would this second message be useful when a tsunami warning would already be sent ? How ? Would the CPA be ready to analyze and use a second message ? What is the national standard operation procedure concerning that issue ? The sensitivity of the parameters (slip and dip variation, rupture zone location and size, and focal depth) should be one of the goal of such method. It is well known that the uncertainty of the magnitude in the first 10 minutes after the quake is about +-0.2. And the focal depth is also not good constrained. The variation of the results with used parameters with the uncertainty should be analyzed.

Referee1 suggested to compare with the DART buoy measurement. As currently, Chile has 6 DART installed along its coast, it would be very useful to compare the amplitude computed by the various models on these 6 DART stations.

The last comment would be on the practical use of such detailed result for a warning purpose. Disaster management authorities need level of warning along the coastlines of their country or county. Typically 3 levels of warning are in place, decided by Unesco : 30 cm, 1m, 3 m. Some countries implemented a 4th level (5 or 6 m). The comparison of run-up computation should take into account such operational criteria to assess the accuracy or the discrepancy between two methods. This should be applied to the set of results. Statistics should be done for the 3 or 4 levels of warning, and for a detailed analysis, it should be demonstrate that the proposed method is more conservative or less conservative than the detailed finer source model. The results of the proposed warning method should be discussed in the scope of the consequences of the difference of warning level with the finest warning level obtained with finer source and finer propagation modeling. Is their method more conservative or less conservative than the finest method ?

Proposed modifications on figures.

a) Figure 2, the scale of the run-up axis should be the same for both figures left and right

b) Figure 5. The presentation of far field and ocean scale results is useless for the Chilean tsunami warning system and not in the scope of this paper. This figure should be removed.

---

## Short Comment (SC1) · 1 Apr 2019

Two remarks which may improve the paper:

(1) I see a logical inconsistency in this Manuscript. On one hand, Authors note that near-field tsunami heights are highly sensitive to slip distribution (e.g., Geist, 2002) and, accordingly, criticize pre-computed scenario databanks for their simplified slip models. On another hand, Authors propose to apply fast W-phase source model for real-time forecasting. But, W-phase CMT is a point source model; there is no slip heterogeneity inside that model. Whether slip distribution is assumed to be uniform, or "elliptical" – makes no difference in sense of source complexity. (One could pre-compute scenario databank with elliptical sources as well). Fast propagation simulations may improve

local early warning only in combination with fast complex source inversion (i.e., slip distribution). Authors do mention FFM modeling on their final flow chart, but there is no any discussion about how they are going to perform fast FFM inversion in 5-10 minutes.

(2) Authors propose linear propagation modeling to forecast tsunami coastal impact. They mention terms "run-ups" and "maximum inundation". However, both run-up and inundation take place on land whereas linear propagation is stopped at 100 m depth. There is no any explanation in the manuscript of how do Authors close the gap between the 100m-isobath and run-ups. Simple Green's law? Such an explanation is very important and must be present in the paper. Same for the non-linear simulations with JAGURS – did they compute the "true runup" over inundated topography (at 15 arc second resolution)? Or extrapolated from some offshore isobath as well? What is then exactly compared between the two models?

———————————————————

---

## Author Comment (AC1) · 11 Apr 2019

Dear Reviewer,

Santiago of Chile, April 11, 2019

We have read carefully your review of our article entitled, "Speeding up and boosting tsunami warning in Chile", written by Fuentes M.(1), Arriola, S. (2), Riquelme S. (2), and Delouis B. (3), from (1) Department of Geophysics, University of Chile, Faculty of Physical and Mathematical Sciences, Santiago, Chile, (2) National Seismological Center, University of Chile, Santiago, Chile and (3) Géoazur, Université de Nice Sophia Antipolis, Observatoire de la Côte d'Azur, Nice, France.

We are grateful for the time you spent to review our paper, for all your comments and

useful suggestions to improve the manuscript. In the following paragraphs we present in detail the answer to all questions, comments and suggestions you made.

Best regards, Mauricio Fuentes.

———————————————————————————————————————————

——— General comments

Reviewer: The paper presents a methodology aimed at speeding up the generation of a tsunami forecast as part of tsunami warning operations. The authors modeled tsunamis for twelve of the largest earthquakes that occurred between 1992 and 2015 applying a) their newly proposed linear method assuming an elliptical slip distribution, and b) a fully non-linear method. Comparison of the results indicates that the proposed linear method allows the generation of a much faster tsunami forecast that matches the results of the fully non-linear method with an accuracy of up to 80% but 20 times faster. These results make the paper worthy of publication and of interest to the tsunami warning and disaster management community. As written, however, the paper needs a significant amount of work before we can consider it ready for publication. The text needs major revisions to improve its overall readability and flow. Instead of providing an exhaustive list of all the grammar and composition issues we have found, however, we have taken the liberty of editing most of the text and will attach it as suggested edits with the hope that it will help with the editing process. Please find below some additional comments and specific suggestions

Response: Thank you very much for the tremendous help you provided us with the edited version with valuable suggestions. We have been incorporated all of them because we think they really improve the manuscript. We provided an annotated version of the manuscript with track of changes (red slanted stands for deleted text and blue for new text.)

———————————————————————————————————————————
———

Specific comments:

(1)

Reviewer: The title of the paper does not reflect the actual contents and results presented in the paper.

The title of the paper suggests that the research results included in it will speed up the issuance of tsunami warnings in Chile. At present, however, most tsunami warning protocols implemented in the world rely on using a quick estimate of an earthquake's magnitude as a proxy to evaluate its tsunamigenic potential. To date, within the context of tsunami warning operations, only the P-wave moment magnitude (Mwp) method implemented at the US Tsunami Warning Centers in the late nineties has significantly speed up tsunami warning in general. More recent earthquake magnitude estimations methods like W-phase, although more robust, accurate, theoretically sound, and faster than other CMT methods, lacked and still lack the speed needed to truly speed up tsunami warning in general. At the time of publication of the seminal paper on the Wphase CMT method paper in 2008, for instance, the PTWC routinely issued tsunami warnings and tsunami messages within 12 minutes of origin time. At present, the PTWC issues its tsunami message products, on average, in less than 6 minutes of origin. In contrast, it still takes between 20-25 minutes to obtain the results of a W-phase CMT inversion, and around 10-15 minutes for a regional implementation. Faster implementations turn possible only in regions with a high density of seismic stations like Chile, Japan, or the West Coast of the United States. Even for these regions the generation an issuance of a tsunami message in less than 5-6 minutes turns close to impossible relying on a W-phase solution. In other words, despite the paper's title, the proposed linear tsunami simulation methodology does not speed up the issuance of tsunami warnings in Chile. The proposed linear method seems to rather speed up considerably the generation of tsunami propagation and inundation forecasts that provide faster and more accurate estimates than those currently in operation. For this reason, the authors should consider changing the title of the paper to something

more reflective of both, the scope of the paper and its results such as: "Speeding up Tsunami Forecasting to boost Tsunami Warning in Chile", with the possible substitution of "boost" with "enhance" or "improve" instead.

Answer: We agree with the observation of the reviewer and we have changed the title of the paper.
* * *
(2)

Reviewer: Towards the end of the introduction the authors state that many of the current warning systems have pre-computed tsunami scenarios at their core. This turns inaccurate, as most warning systems currently operational in the world use the preliminary earthquake location and magnitude as a proxy to evaluate tsunamigenic potential and issue their warnings accordingly. Many use pre-computed tsunami scenarios to generate a tsunami forecast following that initial warning, while others use a combination of precomputed tsunami scenarios and real time tsunami simulations based on the linear shallow water equations. Generation of this last type of forecast currently takes between 3 to 7 seconds for an area covering 1000 to 1500 square kilometers around the earthquake's epicenter, and 10 minutes or less for the whole Pacific basin depending on magnitude and resolution settings. See the reviewed text in the pdf file for suggested edits.

Answer: We have taken the reviewer's comments in order to make the Introduction section clearer.
* * *
(3)

Reviewer: All twelve historical earthquakes used in the study generated tsunamis

recorded by sea-level instruments, either by tide gauges located along the coast or by DART buoys located in deep water. The paper would benefit by the inclusion of a table listing the tsunami waves heights recorded at these point locations together with the corresponding wave heights predicted by both the linear, and non-linear modeling approaches. Doing so would validate not only a model against another considered theoretically superior but also against the actual field measurements of the phenomenon under study. This turns into the ultimate validation of the accuracy and usefulness of both tsunami modeling, and any forecast based on it.

Answer: For comparison purposes, the best option would be use DART bouys, which are free of non-linear effects in open sea. Unfortunately, none of the buoys were enclosed by our computation domains (for near field), except for a Chilean case. Unfortunately, most of the DART stations were deployed in 2015 or 2016. However, we have included the data that we found and added to the manuscript.

——————————————————————————————————————————————————
———

(4)

Reviewer: The authors should consider renaming some of the sections as suggested in the attached pdf file. In addition, the conclusions should list the most relevant results of the study after a brief summary of the work done in the paper. We attempted to summarize the results in the attached pdf file containing a reviewed version of the text, but the authors should consider adding or modifying whatever they consider relevant.

Answer: Thank you very much for the attached suggestion. We have modified the manuscript including those comments. ————————————————————————
————————————————————————————

(5)

Reviewer: The labels of figures and tables should describe their contents to make them

self-contained. When referencing the figures inside the text we suggest applying the same format to all instances, as for instance "Fig. 1", or "Figs. 2 and 3" instead of using "Figure 1". Please find below a list of suggested edits to the current labels of Figures and Tables in the main text. Consider applying similar edits to the labels included in the supplement:

Figure 1. Schematic showing the discretization of the calculation domain for parallel computation.

Figure 2. Near field simulation of the 2015 Illapel earthquake with an elliptical source (left), and a finite fault model (right). The colors assigned to different areas indicate the expected run-ups in meters: a) red for run-ups larger than 3 m, b) orange for runups between 1 and 3 m, c ) yellow for run-ups between 0.3 and 1 m, and d) green for run-ups smaller than 0.3 m.

Figure 3. Regional field simulation of the 2015 Illapel earthquake for an elliptical source (left), and a finite fault model (right). The colors assigned to different areas indicate the expected run-ups in meters: a) red for run-ups larger than 3 m, b) orange for run-ups between 1 and 3 m, c ) yellow for run-ups between 0.3 and 1 m, and d) green for run-ups smaller than 0.3 m.

Figure 4. Normalized run-up energy rate during the first two hours of tsunami simulation. The upper left panel shows the run-up rate along latitude and time, the upper right panel the final maximum run-up, and the bottom left panel the normalized energy rate for the whole process as a time series.

Figure 5. Tsunami travel times across the Pacific basin for the 2015 Illapel earthquake.

Answer: We have changed the text following the reviewer suggestions.

Please also note the supplement to this comment:
https://www.nat-hazards-earth-syst-sci-discuss.net/nhess-2019-9/nhess-2019-9-AC1-

supplement.pdf

[Figure]

**Supplement:**

[revised manuscript text omitted]

**S.1 Elliptic sources data**

**Table S1:** List of W-Phase CMT parameters for the tested events.

| Event | Lat [°] | Lon [°] | Depth [km] | Strike[°] | Dip[°] | Rake [°] | Moment [Nm] | Mw |
|---|---|---|---|---|---|---|---|---|
| 1992 Nicaragua | 11.26 | -87.7258 | 11.5 | 293.1 | 11.7 | 73.5 | 4.55E+27 | 7.71 |
| 2001 Perú | -17.06 | -73.015 | 23.5 | 315.3 | 16.4 | 70.4 | 4.94E+28 | 8.4 |
| 2003 Japan | 42.21 | 143.7746 | 23.5 | 245.6 | 15.6 | 125.3 | 2.21E+28 | 8.16 |
| 2006 Indonesia | -9.284 | 107.419 | 11.5 | 284.1 | 9 | 86.5 | 4.11E+27 | 7.68 |
| 2007 Solomon Isl. | -13.79 | -76.6 | 25.5 | 324.2 | 14 | 63.4 | 2.24E+28 | 8.17 |
| 2007 Perú | -7.86 | 156.3325 | 21.5 | 322.4 | 30 | 101.3 | 1.75E+28 | 8.09 |
| 2009 Samoa | -15.29 | -171.9962 | 15.5 | 306.5 | 22.1 | -115.3 | 1.85E+28 | 8.11 |
| 2010 Chile | -35.95 | -72.71 | 30.5 | 2.3 | 23.5 | 79 | 2.26E+29 | 8.71 |
| 2011 Japan | 37.92 | 143.113 | 19.5 | 196.3 | 11.9 | 85.5 | 4.26E+29 | 9.02 |
| 2012 Canada | 52.47 | -132.13 | 15 | 320 | 29 | 111 | 5.18E+27 | 7.7 |
| 2014 Chile | -19.6097 | -70.7691 | 25 | 328.3 | 16.9 | 70.2 | 2.35E+28 | 8.14 |
| 2015 Chile | -31.38 | -71.77 | 25.5 | 358 | 11.5 | 107.1 | 3.19E+28 | 8.23 |

**S.2 Scenarios tested**

This section provides 24 figures that resumes the tsunami simulation of each event ($\times$ 12) with an elliptic patch and a finite fault model for both, JAGURS code and the simple linear method presented in this work. The left panel shows the source model, yellow triangles are the DART buoys (when available) and white curve is the isobath of 100 m. The central panel displays the maximum tsunami amplitudes with the linear method and above and right are the corresponding run-up heights along longitude and latitude respectively.

[Figure]

**Fig. S1:** Summary of the tsunami simulation for The 1992 Nicaragua Earthquake with an elliptic source.

[Figure]

**Fig. S2:** Summary of the tsunami simulation for The 1992 Nicaragua Earthquake with a finite fault model.

[Figure]

**Fig. S3:** Summary of the tsunami simulation for The 2001 Southern Perú Earthquake with an elliptic source.

[Figure]

**Fig. S4:** Summary of the tsunami simulation for The 2001 Southern Perú Earthquake with a finite fault model.

[Figure]

**Fig. S5:** Summary of the tsunami simulation for The 2003 Japan Earthquake with an elliptic source.

[Figure]

**Fig. S6:** Summary of the tsunami simulation for The 2003 Japan Earthquake with a finite fault model.

[Figure]

**Fig. S7:** Summary of the tsunami simulation for The 2006 Indonesia Earthquake with an elliptic source.

[Figure]

**Fig. S8:** Summary of the tsunami simulation for The 2006 Indonesia Earthquake with a finite fault model.

[Figure]

**Fig. S9:** Summary of the tsunami simulation for The 2007 Solomon Isl. Earthquake with an elliptic source.

[Figure]

**Fig. S10:** Summary of the tsunami simulation for The 2007 Solomon Isl. Earthquake with a finite fault model.

[Figure]

**Fig. S11:** Summary of the tsunami simulation for The 2007 Perú Earthquake with an elliptic source.

[Figure]

**Fig. S12:** Summary of the tsunami simulation for The 2007 Perú Earthquake with a finite fault model.

[Figure]

**Fig. S13:** Summary of the tsunami simulation for The 2009 Samoa Earthquake with an elliptic source.

[Figure]

**Fig. S14:** Summary of the tsunami simulation for The 2009 Samoa Earthquake with a finite fault model.

[Figure]

**Fig. S15:** Summary of the tsunami simulation for The 2010 Chile Earthquake with an elliptic source.

[Figure]

**Fig. S16:** Summary of the tsunami simulation for The 2010 Chile Earthquake with a finite fault model.

[Figure]

**Fig. S17:** Summary of the tsunami simulation for The 2011 Japan Earthquake with an elliptic source.

[Figure]

**Fig. S18:** Summary of the tsunami simulation for The 2011 Japan Earthquake with a finite fault model.

[Figure]

**Fig. S19:** Summary of the tsunami simulation for The 2012 Canada Earthquake with an elliptic source.

[Figure]

**Fig. S20:** Summary of the tsunami simulation for The 2012 Canada Earthquake with a finite fault model.

[Figure]

**Fig. S21:** Summary of the tsunami simulation for The 2014 Chile Earthquake with an elliptic source.

[Figure]

**Fig. S22:** Summary of the tsunami simulation for The 2014 Chile Earthquake with a finite fault model.

[Figure]

**Fig. S23:** Summary of the tsunami simulation for The 2015 Chile Earthquake with an elliptic source.

[Figure]

**Fig. S24:** Summary of the tsunami simulation for The 2015 Chile Earthquake with a finite fault model.

---

## Author Comment (AC2) · 11 Apr 2019

Dear Reviewer,

Santiago of Chile, April 11, 2019

We have read carefully your review of our article entitled, "Speeding up and boosting tsunami warning in Chile", written by Fuentes M.(1), Arriola, S. (2), Riquelme S. (2), and Delouis B. (3), from (1) Department of Geophysics, University of Chile, Faculty of Physical and Mathematical Sciences, Santiago, Chile, (2) National Seismological Center, University of Chile, Santiago, Chile and (3) Géoazur, Université de Nice Sophia Antipolis, Observatoire de la Côte d'Azur, Nice, France.

We are grateful for the time you spent to review our paper, for all your comments and

useful suggestions to improve the manuscript. In the following paragraphs we present in detail the answer to all questions, comments and suggestions you made.

Best regards, Mauricio Fuentes.
* * *
——— General comments

Reviewer: The paper discusses a rapid estimation of potential tsunami energy distribution along the coast (or at certain isobath) from any given earthquake parameters. As a typical forecasting algorithm, the main issue here is the tradeoff between the speed and accuracy. To obtain a timely warning, the proposed algorithm uses a rough source estimate from the W-phase inversion as well as a linear tsunami model. Despite the simplifications, the model produces a sufficient level of accuracy to facilitate the early warning system. Additionally, the proposed method is rigorously tested against historical tsunami events, which is another important factor of this paper that make it worthy of publication. In general, the paper is well-written (except for the discussion and conclusions section) and the main message to convey is easy to follow. However, I would recommend further clarifications in some parts, which can be found in the following specific comments, before the paper can be accepted for publication.

Response: We provided an annotated version of the manuscript with track of changes (red slanted stands for deleted text and blue for new text.) including all your suggestions.
* * *
Specific comments:

(1)

Reviewer: Page 1 Line 8. "Our results show that ... non-linear tsunami code." The sentence can be misleading. I would suggest to revise it into "Our results show that, at

a certain water depth, this linear method : : :".

Answer: We have included this suggestion.
* * ** * *
(2)

Reviewer: Page 2 Line 1. ": : : are based on precomputed scenarios". Adding a sentence here with reference to the previous related works would better justify the statement. Here are some papers that can be considered:

Reymond, D., Okal, E. A., Hébert, H., & Bourdet, M. (2012). Rapid forecast of tsunami wave heights from a database of pre-computed simulations, and application during the 2011 Tohoku tsunami in French Polynesia. Geophysical Research Letters, 39(11).

Gusman, A.R., Tanioka, Y., MacInnes, B.T. & Tsushima, H., 2014. A methodology for near-field tsunami inundation forecasting: Application to the 2011 Tohoku tsunami, J. geophys. Res.: Solid Earth, 119(11), 8186–8206.

Mulia, I. E., Gusman, A. R., & Satake, K., 2018. Alternative to non-linear model for simulating tsunami inundation in real-time. Geophysical Journal International, 214(3), 2002-2013.

Answer: We have included the new references and a sentence to mention them in order to improve the previous statement.
* * ** * *
(3)

Reviewer: Page 3 Equation 1. The first line of the equations is a linearized SWE, and the second line refers to the initial condition. What about the third line? Derivative of

elevations with respect to time at t = 0?

Answer: The first line is the linearized SWE, second and third line are the initial conditions. The second is the initial wave and the third is the equivalent condition for null initial velocity, which is the standard formulation in the static coseismic displacement approach.
* * ** * *
(4)

Reviewer: Page 3 Line 19. The tsunami propagation is limited at 100 m isobath, while in the supporting information the simulated runups are compared with the actual runup observations. I am aware that the paper aims to estimate the possible runup distribution in general by disregarding the physic of nearshore processes due to the nature of the algorithm. However, such an inconsistent comparison needs to be clearly defined. For example, by including the 100 m water depth contour on the plots and additional sentences in the figures caption explaining about the difference of runup locations between observation and model. Or, better yet, why don't use Green's Law as in the Raymond et al. (2012)?

Answer: As the reviewer correctly pointed-out, the linear estimation uses a "linear run-up" estimation which is the case of a reflective vertical wall boundary condition. Roughly, linear and non-linear approaches should be on the same order (Synolakis, 1987; Synolakis, 1991). Certainly, the approach of Reymond et al. (2012) is valid, however we aimed to keep our approach as straightforward as possible, an also, as it was notice by Synolakis (1987), this kind of boundary condition somehow retrieve the Green's law. There is no way to predict detailed run-up heights without a fully coupled non-linear method nor a high-resolution bathymetry, which is out of scope on this work. We have added some sentences to make this clear as well we have added minor modifications in the figures. ———————————————————————
* * *
(5)

Reviewer: Page 5 Table 1. The use of "lon" and "lat" is rather confusing without seeing the corresponding figures. I suggest to add a reference to the supporting information in the Table caption, though it has been mentioned somewhere in the text. Furthermore, mathematical formulation of the correlation coefficient can also be a good addition for the supporting information.

Answer: We have added this reference. —————————————————————————
—————————————————————————————

(6)

Reviewer: Page 6 Table 2. Please explain why the computational time of the elliptic slip distribution is longer than the FFM? From the figures in the supporting information I can see that the elliptic slip models have a smaller subfault size. If that is the case, information on the subfault size should be added in section 2.1 including the reasons for using finer resolution in the elliptic slip model. Also, tT in the caption is written tR on the table.

Answer: The reviewer is right. The size element for the elliptic sources is in general smaller than the FFM. This is to ensure enough resolution on the source model. The typo "tR" was fixed. We have added some sentences making this clear.
* * *
* * *
(7)

Reviewer: Page 6 Line 15. It is difficult to grasp the meaning of the last sentence. Please rewrite it.

Answer: The whole paragraph has been rewritten.
* * *
* * *
(8)

Reviewer: Page 7. The flow of descriptions in the discussion and conclusions section is not very smooth. Improvements can be done by either rewriting the whole paragraphs or using bullet-points or numbers to indicate different topic of discussion.

Answer: The whole section has been rewritten.
* * *
* * *
(9)

Reviewer: For the Java case (Figure S7), it seems like the fault of the elliptic source is located seaward of the trench (in the outer rise region). If this is true, then the model needs to be revised, because the 2006 West Java event was a shallow interplate earthquake (a typical tsunami earthquake), which is better depicted in the FFM solution (Figure S8). Other than that, the Java Island map in the left panel is inaccurate. I believe this may be caused by a wrong color map scale used for plotting. Please also check the other locations.

Answer: Thank you very much for noticing this mistake. This it was a misunderstood when typing the data with a closer event in the same area. We have verified the whole catalog and we have fixed this problem and remaking this scenario and figure.

Please also note the supplement to this comment:
https://www.nat-hazards-earth-syst-sci-discuss.net/nhess-2019-9/nhess-2019-9-AC2-supplement.pdf

————————————————————

2019-9, 2019.

---

## Author Comment (AC3) · 11 Apr 2019

Dear Reviewer,

Santiago of Chile, April 11, 2019

We have read carefully your review of our article entitled, "Speeding up and boosting tsunami warning in Chile", written by Fuentes M.(1), Arriola, S. (2), Riquelme S. (2), and Delouis B. (3), from (1) Department of Geophysics, University of Chile, Faculty of Physical and Mathematical Sciences, Santiago, Chile, (2) National Seismological Center, University of Chile, Santiago, Chile and (3) Géoazur, Université de Nice Sophia Antipolis, Observatoire de la Côte d'Azur, Nice, France.

We are grateful for the time you spent to review our paper, for all your comments and

useful suggestions to improve the manuscript. In the following paragraphs we present in detail the answer to all questions, comments and suggestions you made.

Best regards, Mauricio Fuentes.
* * *
——— General comments

Reviewer: This paper presents a method for quick tsunami estimation in Chile early warning system using W-shape inversion for rough source estimation and linear tsunami numerical modeling. They mention this new approach as a fill-in gap method for the warning system. They have also tested their method with historical tsunami events and proved that they have good correlation between the real case and their results. This work is worth to be published. However, it needs major revision in terms of presenting and discussing their results, grammar in the entire text and conclusion. Besides, it is necessary to have further discussion and explanation on figures.

Response: We provided an annotated version of the manuscript with track of changes (red slanted stands for deleted text and blue for new text.) including all your suggestions.
* * *
* * *
Specific comments:

(1)

Reviewer: Below are the sentences that are not clear to me. They need to be rephrased: 1. Page 1 Line 14 whole sentence 2. Page 2 Line 4 "operating monitoring systems" 3. Page 2 Line 11 "until decreases to 0.5 to 1 m" 4. Page 5 Line 11 : : : whole paragraph 5. Page 5 last paragraph 6. Page 6 Line 4 to 6 7. Page 9 last paragraph.

Below are some of grammar corrections: 1. Page 1 Line 3 "100 km which creates" 2. Page 2 Line 6 "This problem is separated in three parts: the determination of: : :" 3. Page 3 Line 24 ": : :with a fully linear shallow water equation propagation." 4. Page 4 Line 24 ": : :.have tested as many earthquakes listed below and: : :" 5. Page 6 Line 2 ": : :very first minutes: : :" 6. Page 6 Line 11 ": : :which deploy a specific: : :" 7. Page 7 Line 1 "These kind of maps: : :" 8. Page 7 Line 5 "with a unique and simple: : :" 9. Page 7 Line 9 ": : :and number of people exposed to this hazard: : :" 10. Page 7 Line 13 ": : :details rather than modeling: : :" 11. Page 10 Line 1 "Using the methodology of Sandabata el al. (2018): : :"

Answer: We have rephrased the observed parts and fixed the English grammar.
* * *
(2)

Reviewer: About Figures and Tables: 1. Please refer to Figure 1 in the text. 2. please insert Table 1 after its reference in the text. 3. Figure 2: Does this red color on land represent runup values larger than 3m? Does this mean all of this are experienced more than 3m runup?? 4. Figure 5: please compare and discuss the difference between two maps: in terms of the effect of dispersion etc.. 5. Figure 6: please refer Figure 6 in the text and explain.

Answer:

1.) Figure 1 is now referred in section 3.

2.- ) Locating the figures through the manuscript seems to be a post editorial task, since we can not correctly control the insertion of the figures with the provided LaTeX template. However, every reference to a figure or table is mentioned before they are inserted (in the .tex file).

3.-) In figure 2 (and 3), the coastline is divided by geopolitical zones (Chilean regions),

the zone adopts the color of the maximum value of the runup distribution in that zone. If only one point overpass 3 m, the whole region becomes red. That is why we pointed-out that this way to divide the country is just referential, because we can easily use another, not being relevant for the algorithm development and more related to the criteria of the final user.

4.) The caption of figure 5 was rewritten in order to make this clear.

5.) Figure 6 is now referred in section 6.

Please also note the supplement to this comment:
https://www.nat-hazards-earth-syst-sci-discuss.net/nhess-2019-9/nhess-2019-9-AC3-supplement.pdf

---

## Author Comment (AC4) · 11 Apr 2019

Dear Reviewer,

Santiago of Chile, April 11, 2019

We have read carefully your review of our article entitled, "Speeding up and boosting tsunami warning in Chile", written by Fuentes M.(1), Arriola, S. (2), Riquelme S. (2), and Delouis B. (3), from (1) Department of Geophysics, University of Chile, Faculty of Physical and Mathematical Sciences, Santiago, Chile, (2) National Seismological Center, University of Chile, Santiago, Chile and (3) Géoazur, Université de Nice Sophia Antipolis, Observatoire de la Côte d'Azur, Nice, France.

We are grateful for the time you spent to review our paper, for all your comments and

useful suggestions to improve the manuscript. In the following paragraphs we present in detail the answer to all questions, comments and suggestions you made.

Best regards, Mauricio Fuentes.

————————————————————————————————————————————

——— General comments

Reviewer: The paper presents a methodology to speeding up the tsunami forecast in Chile as part of tsunami warning operations. This is a very important topic, in particular because in the last decade many new tsunami warning centers have been established by various countries. This paper presents interesting results for publication. Nevertheless, several issues should be explained, discussed and many data are missing, before accepting the paper. Major revision is necessary.

Response: We provided an annotated version of the manuscript with track of changes (red slanted stands for deleted text and blue for new text.) including all your suggestions.

————————————————————————————————————————————

—————

Specific comments:

(1)

Reviewer: The results of proposed methods depends mainly on the variation of the source parameters between the different methods used, in particular the slip, the dip and the dimension and location of rupture zone, and the focal depth. One first request : the list and values of parameters of the sources of that paper are missing.

Answer: We added a new table in the supplemental material containing the requested information.

————————————————————————————————————————————

————

(2)

Reviewer: It can already been checked on the various maps presented, that the location of the epicenter for the elliptic model, and the location of the center of the rupture zone of the fault model are not the same, and there are not the GCMT location. Why ? How do the authors decide the location of the epicenter, and why the locations are different for the different models ?

Answer: Thank you for noticing this mistake. In the elliptical model, the star stands for Centroid location whereas in the FFM model, the star denotes the epicenter location. We have fixed this in the figures and changed the symbol for the centroid in order to avoid confusion.
* * *
————

(3)

Reviewer: The second question is why did the authors analyzed the results of such method along other coastlines than Chile? It doesn't provide any results about the variability of the warning forecast along the Chilean coastlines. On the other hand, two missing recent events have not been modeled and should be added to the study: Chile 1985 and Antofagasta 1995.

Answer: The main reason is to validate the linear method for the propagation of the tsunami, which needs to be tested in different scenarios. Once we have certain degree of confidence (in statistical terms), we apply it to the particular case of Chile, but not being excluding to be useful in other regions. Also, we decided to pick the last three Chilean tsunamis with associated moment magnitude bigger that 8.0, which also are well recorded and documented.
* * ** * *
(4)

Reviewer: One of the recent papers that describes the effectiveness and rapidity of the W-Phase to get robust centroid moment tensor solutions is by J. Roch at al. (Roch, J., Duperray, P. and Schindelé F. (2016) Very fast characterization of focal mechanism parameters through W-Phase Centroid inversion in the context of tsunami warning, Pure Appl. Geophys. 173 (2016), 3881–3893, DOI 10.1007/s00024-016-1258-3). In that paper, the authors analyzed W-Phase results at global and regional scale with specific Green's functions to provide accurate solution in 15 minutes (10 minutes of signal). Due to the characteristics of the very long period W-Phase, it wouldn't be physically feasible to compute sooner W-Phase waves. But it is well known that the first tsunami wave could impact the Chilean coastlines in less than 15 minutes. The mandate and goal of the National tsunami warning center that is facing near-field tsunami warning is to provide the first warning message in less than 15 minutes after the quake occurrence. As the results of W-Phase would not be available, the authors should explain how they would proceed to provide this first bulletin. The authors should identify a preliminary solution to perform modeling before getting the results of the W-Phase computation and getting results in 15 minutes after the quake.

Answer:

In Zhao et. al. 2017 and Riquelme et.al 2018, the possibility to have a W-phase CMT in 6 minutes is studied with good results. Thus, the fact of computing a W-phase solution in a very short time after earthquake location is well reported. We provide you the doi of both papers.

Zhao et al: https://doi.org/10.1002/2017JB014950 Riquelme et al: https://doi.org/10.1785/0220180146
* * ** * *
(5)

Reviewer: Second point, the authors informed that for their study, they used W-Phase results. How was computed the parameters of all these past events ? In particular, the location of the centroid moment tensor, and on the strike and dip values used for the elliptic method. On this specific method, the authors should present what are the parameters of the seismic sources needed for the elliptic method, and the values of the parameters for all the events processed in this paper.

Answer: The W-phase method provides the full moment tensor. We just retrieved data from the National Seismological Center in Chile. The data can be accessed through IRIS (www.iris.edu) or USGS (https://earthquake.usgs.gov/). For instance, the W-phae solution of the Nicaragua Earthquake (the oldest in our list) is here: https://earthquake.usgs.gov/earthquakes/eventpage/usp0005ddn/moment-tensor

Therefore, all the parameters you mention are given by this method. Also, we have added a table in the supplemental material with the values used for the elliptic models.
* * ** * *
(6)

Reviewer: The next issue is how they plan to implement the complementary messages using W-Phase source parameters. Would this second message be useful when a tsunami warning would already be sent ? How ? Would the CPA be ready to analyze and use a second message ? What is the national standard operation procedure concerning that issue ? The sensitivity of the parameters (slip and dip variation, rupture zone location and size, and focal depth) should be one of the goal of such method. It is well known that the uncertainty of the magnitude in the first 10 minutes after the quake is about +-0.2. And the focal depth is also not good constrained. The variation of the

results with used parameters with the uncertainty should be analyzed. Referee1 suggested to compare with the DART buoy measurement. As currently, Chile has 6 DART installed along its coast, it would be very useful to compare the amplitude computed by the various models on these 6 DART stations.

Answer: Despite there is uncertainty in each of the parameters, we don't try to solve that issue in this paper, but to show how a simple linear method can dramatically decrease the computation times keeping a high degree of accuracy, when compared with standard non-linear methods and the potential for early warning purposes. Nevertheless, the things that the reviewer pointed-out are of high interest and deserve a dedicated study. We have addressed those comments in the discussion section as a future work, including about DART buoys. However, several of the DART stations in Chilean coasts were deployed in 2015-2016 not being possible to use all of them in this study. Also the majority of the buoys belongs to the "far-field domain".
* * *
(7)

Reviewer: The last comment would be on the practical use of such detailed result for a warning purpose. Disaster management authorities need level of warning along the coastlines of their country or county. Typically, 3 levels of warning are in place, decided by Unesco: 30 cm, 1m, 3 m. Some countries implemented a 4th level (5 or 6 m). The comparison of run-up computation should take into account such operational criteria to assess the accuracy or the discrepancy between two methods. This should be applied to the set of results. Statistics should be done for the 3 or 4 levels of warning, and for a detailed analysis, it should be demonstrated that the proposed method is more conservative or less conservative than the detailed finer source model. The results of the proposed warning method should be discussed in the scope of the consequences of the difference of warning level with the finest warning level obtained with finer source

and finer propagation modeling. Is their method more conservative or less conservative than the finest method?

Answer: Once the method is developed, the final user can decide what geopolitical subdivision is more suitable, as well as the number of warning levels. Both are easily adjustable in the methodology being part of the criteria adopted for the government institutions. It is hard to say which one could be more conservative even with such statistical analysis, because there are other factors, namely psychological, communicational, etc. One should have a good compromise between quantitative results and simplicity on the way the information is transmitted (which can be somehow subjective). Nonetheless, this discussion is highly valuable, and we have included in the manuscript.
* * *
* * *
(8)

Reviewer: Proposed modifications on figures.

a) Figure 2, the scale of the run-up axis should be the same for both figures left and right

b) Figure 5. The presentation of far field and ocean scale results is useless for the Chilean tsunami warning system and not in the scope of this paper. This figure should be removed.

Answer:

a) Figure 2 is now with same scales.

b) One of the main objectives of this work is to show the power of the linear method, so another simple and fast application, is to compliment any scenario with a global map of travel times, even allowing the inclusion of different effects.

Please also note the supplement to this comment:
https://www.nat-hazards-earth-syst-sci-discuss.net/nhess-2019-9/nhess-2019-9-AC4-supplement.pdf

**Supplement:**

[revised manuscript text omitted]

**S.1 Elliptic sources data**

**Table S1:** List of W-Phase CMT parameters for the tested events.

| Event | Lat [°] | Lon [°] | Depth [km] | Strike[°] | Dip[°] | Rake [°] | Moment [Nm] | Mw |
|---|---|---|---|---|---|---|---|---|
| 1992 Nicaragua | 11.26 | -87.7258 | 11.5 | 293.1 | 11.7 | 73.5 | 4.55E+27 | 7.71 |
| 2001 Perú | -17.06 | -73.015 | 23.5 | 315.3 | 16.4 | 70.4 | 4.94E+28 | 8.4 |
| 2003 Japan | 42.21 | 143.7746 | 23.5 | 245.6 | 15.6 | 125.3 | 2.21E+28 | 8.16 |
| 2006 Indonesia | -9.284 | 107.419 | 11.5 | 284.1 | 9 | 86.5 | 4.11E+27 | 7.68 |
| 2007 Solomon Isl. | -13.79 | -76.6 | 25.5 | 324.2 | 14 | 63.4 | 2.24E+28 | 8.17 |
| 2007 Perú | -7.86 | 156.3325 | 21.5 | 322.4 | 30 | 101.3 | 1.75E+28 | 8.09 |
| 2009 Samoa | -15.29 | -171.9962 | 15.5 | 306.5 | 22.1 | -115.3 | 1.85E+28 | 8.11 |
| 2010 Chile | -35.95 | -72.71 | 30.5 | 2.3 | 23.5 | 79 | 2.26E+29 | 8.71 |
| 2011 Japan | 37.92 | 143.113 | 19.5 | 196.3 | 11.9 | 85.5 | 4.26E+29 | 9.02 |
| 2012 Canada | 52.47 | -132.13 | 15 | 320 | 29 | 111 | 5.18E+27 | 7.7 |
| 2014 Chile | -19.6097 | -70.7691 | 25 | 328.3 | 16.9 | 70.2 | 2.35E+28 | 8.14 |
| 2015 Chile | -31.38 | -71.77 | 25.5 | 358 | 11.5 | 107.1 | 3.19E+28 | 8.23 |

**S.2 Scenarios tested**

This section provides 24 figures that resumes the tsunami simulation of each event ($\times$ 12) with an elliptic patch and a finite fault model for both, JAGURS code and the simple linear method presented in this work. The left panel shows the source model, yellow triangles are the DART buoys (when available) and white curve is the isobath of 100 m. The central panel displays the maximum tsunami amplitudes with the linear method and above and right are the corresponding run-up heights along longitude and latitude respectively.

[Figure]

**Fig. S1:** Summary of the tsunami simulation for The 1992 Nicaragua Earthquake with an elliptic source.

[Figure]

**Fig. S2:** Summary of the tsunami simulation for The 1992 Nicaragua Earthquake with a finite fault model.

[Figure]

**Fig. S3:** Summary of the tsunami simulation for The 2001 Southern Perú Earthquake with an elliptic source.

[Figure]

**Fig. S4:** Summary of the tsunami simulation for The 2001 Southern Perú Earthquake with a finite fault model.

[Figure]

**Fig. S5:** Summary of the tsunami simulation for The 2003 Japan Earthquake with an elliptic source.

[Figure]

**Fig. S6:** Summary of the tsunami simulation for The 2003 Japan Earthquake with a finite fault model.

[Figure]

**Fig. S7:** Summary of the tsunami simulation for The 2006 Indonesia Earthquake with an elliptic source.

[Figure]

**Fig. S8:** Summary of the tsunami simulation for The 2006 Indonesia Earthquake with a finite fault model.

[Figure]

**Fig. S9:** Summary of the tsunami simulation for The 2007 Solomon Isl. Earthquake with an elliptic source.

[Figure]

**Fig. S10:** Summary of the tsunami simulation for The 2007 Solomon Isl. Earthquake with a finite fault model.

[Figure]

**Fig. S11:** Summary of the tsunami simulation for The 2007 Perú Earthquake with an elliptic source.

[Figure]

**Fig. S12:** Summary of the tsunami simulation for The 2007 Perú Earthquake with a finite fault model.

[Figure]

**Fig. S13:** Summary of the tsunami simulation for The 2009 Samoa Earthquake with an elliptic source.

[Figure]

**Fig. S14:** Summary of the tsunami simulation for The 2009 Samoa Earthquake with a finite fault model.

[Figure]

**Fig. S15:** Summary of the tsunami simulation for The 2010 Chile Earthquake with an elliptic source.

[Figure]

**Fig. S16:** Summary of the tsunami simulation for The 2010 Chile Earthquake with a finite fault model.

[Figure]

**Fig. S17:** Summary of the tsunami simulation for The 2011 Japan Earthquake with an elliptic source.

[Figure]

**Fig. S18:** Summary of the tsunami simulation for The 2011 Japan Earthquake with a finite fault model.

[Figure]

**Fig. S19:** Summary of the tsunami simulation for The 2012 Canada Earthquake with an elliptic source.

[Figure]

**Fig. S20:** Summary of the tsunami simulation for The 2012 Canada Earthquake with a finite fault model.

[Figure]

**Fig. S21:** Summary of the tsunami simulation for The 2014 Chile Earthquake with an elliptic source.

[Figure]

**Fig. S22:** Summary of the tsunami simulation for The 2014 Chile Earthquake with a finite fault model.

[Figure]

**Fig. S23:** Summary of the tsunami simulation for The 2015 Chile Earthquake with an elliptic source.

[Figure]

**Fig. S24:** Summary of the tsunami simulation for The 2015 Chile Earthquake with a finite fault model.

---

## Author Comment (AC5) · 11 Apr 2019

Dear Dr Babeyko,

Santiago of Chile, April 00, 2019

We have read carefully your review of our article entitled, "Speeding up and boosting tsunami warning in Chile", written by Fuentes M.(1), Arriola, S. (2), Riquelme S. (2), and Delouis B. (3), from (1) Department of Geophysics, University of Chile, Faculty of Physical and Mathematical Sciences, Santiago, Chile, (2) National Seismological Center, University of Chile, Santiago, Chile and (3) Géoazur, Université de Nice Sophia Antipolis, Observatoire de la Côte d'Azur, Nice, France.

We are grateful for the time you spent to review our paper, for all your comments and

useful suggestions to improve the manuscript. In the following paragraphs we present in detail the answer to all questions, comments and suggestions you made.

Best regards, Mauricio Fuentes.
* * *
——— General comments

Reviewer:

Two remarks which may improve the paper:

(1) I see a logical inconsistency in this Manuscript. On one hand, Authors note that near-field tsunami heights are highly sensitive to slip distribution (e.g., Geist, 2002) and, accordingly, criticize pre-computed scenario databanks for their simplified slip models. On another hand, Authors propose to apply fast W-phase source model for real-time forecasting. But, W-phase CMT is a point source model; there is no slip heterogeneity inside that model. Whether slip distribution is assumed to be uniform, or "elliptical" – makes no difference in sense of source complexity. (One could pre-compute scenario databank with elliptical sources as well). Fast propagation simulations may improve local early warning only in combination with fast complex source inversion (i.e., slip distribution). Authors do mention FFM modeling on their final flow chart, but there is no any discussion about how they are going to perform fast FFM inversion in 5-10 minutes.

(2) Authors propose linear propagation modeling to forecast tsunami coastal impact. They mention terms "run-ups" and "maximum inundation". However, both run-up and inundation take place on land whereas linear propagation is stopped at 100 m depth. There is no any explanation in the manuscript of how do Authors close the gap between the 100m-isobath and run-ups. Simple Green's law? Such an explanation is very important and must be present in the paper. Same for the non-linear simulations with JAGURS – did they compute the "true runup" over inundated topography (at 15 arc second resolution)? Or extrapolated from some offshore isobath as well? What is then

exactly compared between the two models?

Answer:

(1) It is a good observation. That is the reason we try to give one step in-between the uniform slip models and FFM which are not currently available in real time. Elliptical models for the same magnitude, allows to concentrate more slip in a given region than the uniform slip, which turns this elliptical model "worse" (in terms of threat) than the uniform one.

The scope of this work is on the power of the linear solution, on the reference list you will find the articles that decribe the "why and how" of obtaining FFM models within 5 min.

(2) We have made a new version of the manuscript that better explains this point. However, the problem is the same for both, linear and non-linear since we don't have fine bathymetry data to nest grids and "truly" compute the run-up/inundation . The approximated run-up is just the peak that is registered in the vertical wall (boundary condition). If users would want to use finer bathymetry, the ideas proposed here will be the same.

Please also note the supplement to this comment:
https://www.nat-hazards-earth-syst-sci-discuss.net/nhess-2019-9/nhess-2019-9-AC5-supplement.pdf

**Supplement:**

[revised manuscript text omitted]

**S.1 Elliptic sources data**

**Table S1:** List of W-Phase CMT parameters for the tested events.

| Event | Lat [°] | Lon [°] | Depth [km] | Strike[°] | Dip[°] | Rake [°] | Moment [Nm] | Mw |
|---|---|---|---|---|---|---|---|---|
| 1992 Nicaragua | 11.26 | -87.7258 | 11.5 | 293.1 | 11.7 | 73.5 | 4.55E+27 | 7.71 |
| 2001 Perú | -17.06 | -73.015 | 23.5 | 315.3 | 16.4 | 70.4 | 4.94E+28 | 8.4 |
| 2003 Japan | 42.21 | 143.7746 | 23.5 | 245.6 | 15.6 | 125.3 | 2.21E+28 | 8.16 |
| 2006 Indonesia | -9.284 | 107.419 | 11.5 | 284.1 | 9 | 86.5 | 4.11E+27 | 7.68 |
| 2007 Solomon Isl. | -13.79 | -76.6 | 25.5 | 324.2 | 14 | 63.4 | 2.24E+28 | 8.17 |
| 2007 Perú | -7.86 | 156.3325 | 21.5 | 322.4 | 30 | 101.3 | 1.75E+28 | 8.09 |
| 2009 Samoa | -15.29 | -171.9962 | 15.5 | 306.5 | 22.1 | -115.3 | 1.85E+28 | 8.11 |
| 2010 Chile | -35.95 | -72.71 | 30.5 | 2.3 | 23.5 | 79 | 2.26E+29 | 8.71 |
| 2011 Japan | 37.92 | 143.113 | 19.5 | 196.3 | 11.9 | 85.5 | 4.26E+29 | 9.02 |
| 2012 Canada | 52.47 | -132.13 | 15 | 320 | 29 | 111 | 5.18E+27 | 7.7 |
| 2014 Chile | -19.6097 | -70.7691 | 25 | 328.3 | 16.9 | 70.2 | 2.35E+28 | 8.14 |
| 2015 Chile | -31.38 | -71.77 | 25.5 | 358 | 11.5 | 107.1 | 3.19E+28 | 8.23 |

**S.2 Scenarios tested**

This section provides 24 figures that resumes the tsunami simulation of each event ($\times$ 12) with an elliptic patch and a finite fault model for both, JAGURS code and the simple linear method presented in this work. The left panel shows the source model, yellow triangles are the DART buoys (when available) and white curve is the isobath of 100 m. The central panel displays the maximum tsunami amplitudes with the linear method and above and right are the corresponding run-up heights along longitude and latitude respectively.

[Figure]

**Fig. S1:** Summary of the tsunami simulation for The 1992 Nicaragua Earthquake with an elliptic source.

[Figure]

**Fig. S2:** Summary of the tsunami simulation for The 1992 Nicaragua Earthquake with a finite fault model.

[Figure]

**Fig. S3:** Summary of the tsunami simulation for The 2001 Southern Perú Earthquake with an elliptic source.

[Figure]

**Fig. S4:** Summary of the tsunami simulation for The 2001 Southern Perú Earthquake with a finite fault model.

[Figure]

**Fig. S5:** Summary of the tsunami simulation for The 2003 Japan Earthquake with an elliptic source.

[Figure]

**Fig. S6:** Summary of the tsunami simulation for The 2003 Japan Earthquake with a finite fault model.

[Figure]

**Fig. S7:** Summary of the tsunami simulation for The 2006 Indonesia Earthquake with an elliptic source.

[Figure]

**Fig. S8:** Summary of the tsunami simulation for The 2006 Indonesia Earthquake with a finite fault model.

[Figure]

**Fig. S9:** Summary of the tsunami simulation for The 2007 Solomon Isl. Earthquake with an elliptic source.

[Figure]

**Fig. S10:** Summary of the tsunami simulation for The 2007 Solomon Isl. Earthquake with a finite fault model.

[Figure]

**Fig. S11:** Summary of the tsunami simulation for The 2007 Perú Earthquake with an elliptic source.

[Figure]

**Fig. S12:** Summary of the tsunami simulation for The 2007 Perú Earthquake with a finite fault model.

[Figure]

**Fig. S13:** Summary of the tsunami simulation for The 2009 Samoa Earthquake with an elliptic source.

[Figure]

**Fig. S14:** Summary of the tsunami simulation for The 2009 Samoa Earthquake with a finite fault model.

[Figure]

**Fig. S15:** Summary of the tsunami simulation for The 2010 Chile Earthquake with an elliptic source.

[Figure]

**Fig. S16:** Summary of the tsunami simulation for The 2010 Chile Earthquake with a finite fault model.

[Figure]

**Fig. S17:** Summary of the tsunami simulation for The 2011 Japan Earthquake with an elliptic source.

[Figure]

**Fig. S18:** Summary of the tsunami simulation for The 2011 Japan Earthquake with a finite fault model.

[Figure]

**Fig. S19:** Summary of the tsunami simulation for The 2012 Canada Earthquake with an elliptic source.

[Figure]

**Fig. S20:** Summary of the tsunami simulation for The 2012 Canada Earthquake with a finite fault model.

[Figure]

**Fig. S21:** Summary of the tsunami simulation for The 2014 Chile Earthquake with an elliptic source.

[Figure]

**Fig. S22:** Summary of the tsunami simulation for The 2014 Chile Earthquake with a finite fault model.

[Figure]

**Fig. S23:** Summary of the tsunami simulation for The 2015 Chile Earthquake with an elliptic source.

[Figure]

**Fig. S24:** Summary of the tsunami simulation for The 2015 Chile Earthquake with a finite fault model.

---

## Author Response (AR2)

Dear Reviewer,

We have read carefully your review of our article entitled, "Speeding up and boosting tsunami warning in Chile", written by Fuentes M.[1], Arriola, S. [2], Riquelme S. [2], and Delouis B. [3], from (1) Department of Geophysics, University of Chile, Faculty of Physical and Mathematical Sciences, Santiago, Chile, (2) National Seismological Center, University of Chile, Santiago, Chile and (3) Géoazur, Université de Nice Sophia Antipolis, Observatoire de la Côte d'Azur, Nice, France.

We are grateful for the time you spent to review our paper, for all your comments and useful suggestions to improve the manuscript. In the following paragraphs we present in detail the answer to all questions, comments and suggestions you made.

Best regards,
Mauricio Fuentes.
* * *
General comments

**Reviewer:**  Please make the following corrections before publication
On Page 2, Line 1, the text reads "it allows to obtain a moment tensor solution within 5 minutes". In my opinion the authors should clarify whether they refer to an actual capability already made operational as part of the Chilean tsunami warning system, or to the not-yet-realized possibilities of the Wphase CMT method. Listing some operational examples would also help legitimize the 5 minutes claim. Such Wphase CMT processing speed might currently turn possible only in a few regions of the world, including Chile, given the higher level of instrumentation needed to achieve such results. My comment has to do with the fact that in our experience there exist a significant difference between research done in academia and the actual implementation of a method or technique as part of a robust 24/7 monitoring system. If the statement regarding the attainment of a moment tensor solution within 5 minutes applies to the Chilean tsunami warning system currently in operation, then the paper would also benefit from some clarification of whether it uses seismic, GPS data, or a combination of both.

**Response:**  Thank you very much for your observations. He have clarified this point regarding the W-phase in the text and also including a new reference.
* * *
Specific comments:

**Reviewer:**

Page 2, Line 2: "...however, that tsunami heigts are..."
→ ...however, that tsunami wave heights are...

Page 2, Line 13: "...data with 15 arcsec of resolution..."
→ ...data with a 15 arcsec resolution...

Page 2, Line 23: "...determined with the scaling laws obtained by Blaser et al. (2010)."
→ ...determined by applying the scaling laws after Blaser et al. (2010).

Page 2, Line 24: "With ny = 16 the studied cases have enough resolution on the source area".
→ After setting ny = 16 all the earthquake cases analyzed in our study have enough resolution on the source area.

Page 2, Line 30: "...as sugested in Tanioka...
→ ...as suggested by Tanioka...

Page 3, Line 10: "...in a vertical wall placed at an isobath of 100m,..."
→ ...in a vertical wall placed at the 100 m isobath,...

Page 3, Line 12: "...to obtain a quicker..."
→ ...to obtain a faster...

Page 4, Line 1: "...on the classical finite..."
→ ...on the classic finite...

Page 4, Line 4: "This is usually on the same order with the actual runup in..."
→ The resulting runup values are in the same order of the actual runups for...

Page 4, Line 10: "...have proven to be operationally..."
→ ...have proven operationally...

Page 4, Line 27: "The extension of the earthquakes..."
→ The geometry of the earthquakes' causative falts...

Page 7, Line 10: "This makes sense since...
→ This makes sense, since...

Page 9, Line 8: "...equations to model more than 80% of the runups..."

→ ...equations. Implementation of this method allows to model more than 80% of the tsunami runups

Page 10, Line 1: "...of the runups using..."
→ ...of the tsunami runups using…

Page 5, Line 4: "...in a flow chart (Fig. 6)."
→ ...in the flowchart shown in Figure 6.

Page 5, Line 8: "...a fast runup estimation..."
→...a fast tsunami runup estimation…

Page 5, Line 9: "...time we estimate…
→ ...time we can estimate…

**Answer:** We have included all these suggestions.